# Hippocampal convergence during anticipatory midbrain activation promotes subsequent memory formation

Jia-Hou Poh [1] ✉, Mai-Anh T. Vu[1,2,3], Jessica K. Stanek[1,4], Abigail Hsiung[1,4], Tobias Egner [1,4] & R. Alison Adcock [1,2,4,5] ✉

The hippocampus has been a focus of memory research since H.M's surgery abolished his ability to form new memories, yet its mechanistic role in memory remains debated. Here, we identify a candidate memory mechanism: an anticipatory hippocampal "convergence state", observed while awaiting valuable information, and which predicts subsequent learning. During fMRI, participants viewed trivia questions eliciting high or low curiosity, followed seconds later by its answer. We reasoned that encoding success requires a confluence of conditions, so that hippocampal states more conducive to memory formation should converge in state space. To operationalize convergence of neural states, we quantified the typicality of multivoxel patterns in the medial temporal lobes during anticipation and encoding of trivia answers. We found that the typicality of anticipatory hippocampal patterns increased during high curiosity. Crucially, anticipatory hippocampal pattern typicality increased with dopaminergic midbrain activation and uniquely accounted for the association between midbrain activation and subsequent recall. We propose that hippocampal convergence states may complete a cascade from motivation and midbrain activation to memory enhancement, and may be a general predictor of memory formation.

The mysterious translation of daily life into the faulty record of memory has long compelled human wonder and conflict. A growing literature has shown that, while some kinds of events are inherently more memorable, our ongoing motivational states are important determinants of whether and how experience is remembered[1–6]. It has been known for decades that neurotransmitters associated with motivation influence neural plasticity at the cellular level[7–9]. More recent research has identified network relationships between nuclei that release these neurotransmitters and the hippocampus, long implicated in memory formation, to show that these relationships predict memory formation[10,11]. These studies have traced a cascade from motivational state, to engagement of neuromodulatory nuclei, to activation of hippocampal systems, concluding *that* neuromodulation

helps the hippocampus create memories. However, they do not answer the question of how activation in neuromodulatory nuclei might create hippocampal states conducive for memory formation. This is perhaps unsurprising given that, despite its remarkable anatomy and physiological specializations, researchers still debate the canonical function of the hippocampus and its role in memory[12–16].

The mesolimbic dopamine system is the major neuromodulatory system implicated in motivated learning of valuable information[17]. It has been proposed that the motivation to learn engages mesolimbic dopaminergic circuits to support plasticity in the hippocampus[18,19]. Consistent with this account, fMRI studies in humans have shown that the motivation to learn, inspired by both intrinsic and extrinsic rewards, is accompanied by increased anticipatory activation in the

[1]Center for Cognitive Neuroscience, Duke University, Durham, NC, USA. [2]Department of Neurobiology, Duke University, Durham, NC, USA. [3]Department of Psychological & Brain Sciences, Boston University, Boston, MA, USA. [4]Department of Psychology & Neuroscience, Duke University, Durham, NC, USA. [5]Department of Psychiatry & Behavioral Sciences, Duke University, Durham, NC, USA. ✉e-mail: jiahou.poh@duke.edu; alison.adcock@duke.edu

dopaminergic midbrain[18]. Such increases in midbrain BOLD have been shown to correlate with PET measures of dopamine release in target regions[20]. During anticipatory midbrain activation, greater functional connectivity between the midbrain and regions in the medial temporal lobe, including the hippocampus (HPC)[18,19,21], predicts memory. Intriguingly, in one such study where participants were motivated by intrinsic curiosity, memory was enhanced not only for the information of interest, but also for temporally proximal irrelevant information[19], suggesting a sustained state of enhanced encoding.

Independent lines of work suggest that the hippocampus can express different functional states that reflect neuromodulation, and that these may manifest as physiological signatures associated with distinct patterns of activity in the hippocampus. First, the hippocampus receives inputs from the dopaminergic midbrain, including ventral tegmental area (VTA)[22–24], and midbrain projections modulate hippocampal physiology and influence performance on memory tasks[25–29]. Studies of place cells in rodents have also shown that place field stability is influenced by task goals, dependent on midbrain modulation[1,30]. Such shifts in response properties and circuit function have physiological signatures that could manifest in BOLD activation patterns[31]. Indeed, in humans, dopamine receptor density has been associated with variability in the BOLD signal intensity in the hippocampus[32]. The anatomical separation of mesolimbic terminals relative to dopamine receptors in the hippocampus is ill-suited to temporally precise signals[17] and further suggests that midbrain dopamine regulates expression of sustained functional states in the hippocampus conducive to encoding[1,17,30,33].

Second, memory-related patterns of activity in the hippocampus, detectable using multivoxel pattern analysis (MVPA) of fMRI data[34–37] have been shown to reflect incentive contexts and goals. With MVPA, patterns of activity across spatially distributed voxels can be formulated as points within a high dimensional state space, with activity in each voxel constituting a single dimension. Multivoxel patterns in the hippocampus differentiate reward contexts and predict individual differences in reward-related memory benefits[34]. When selective attention was manipulated by changing task goals, stability of hippocampal representations for the goal category predicted memory[36]. In addition to representations of rewarded contexts and goal-relevant stimuli, multivoxel patterns in the hippocampus have also been shown to differentiate processes associated with encoding or retrieval[38]. Thus, while variations in the hippocampal patterns are generally studied as informational representations, these findings and others suggest that hippocampal patterns and states may also manifest the effects of neuromodulation.

Here, we were motivated by the premise that hippocampal multivoxel activation patterns may manifest, in addition to representations of information, neural states conducive to memory formation. We reasoned that if conducive state-spaces exist, their instantiation would be a candidate mechanism for memory enhancement by neuromodulation. To isolate patterns associated with neuromodulatory effects from those representing information to be encoded, we examined intervals prior to the presentation of memoranda. We hypothesized that while awaiting valuable information, (i) hippocampal states would reflect the univariate activation of dopaminergic midbrain VTA, (ii) that instantiation of patterns associated with memory-conducive neural states would predict subsequent memory, and (iii) that memory-conducive states would account for previously reported associations between increased midbrain VTA activation and memory formation.

We posited that hippocampal states associated with successful memory formation would fall within an optimally conducive subspace. To identify the optimal subspace, we can exploit the likelihood that the successful formation of new memories requires the convergence of multiple cognitive and physiological factors. If so, the lack of any factor could impede memory formation, yielding multiple ways to fail. An intuition for this *convergence* state principle is captured in the opening of Leo Tolstoy's novel Anna Karenina - "All happy families are alike; each unhappy family is unhappy in its own way". The "Anna Karenina principle" has been applied to the study of dynamical systems (e.g., ref. [39]), including the examination of human brain networks[40]. Related lines of work on spontaneous brain dynamics have consistently associated better perceptual processing with reduced neural variability in sensory cortices[41,42]. Here, we apply similar logic to identification of brain states conducive to successful memory formation.

Our proposed usage of *convergence state* denotes the manifestation of an optimal subspace among neural states when multiple conditions are met. While building on and sharing conceptual commonalities with prior formulations of the hippocampus as a convergence zone where multiple streams of information are integrated (e.g., refs. [43–47]), we use *convergence states* to indicate not only an anatomical nexus, but also co-occurrence of conditions that support the encoding of information. These conditions are not per se representations of information; they may relate to processes described using words like arousal, attention, or neuromodulation. To the extent that midbrain VTA neuromodulation engages a hippocampal state conducive for memory formation, the convergence state principle predicts that, compared to states associated with failed encoding, these neural convergence states would be closer to a prototypical state. Thus, we hypothesized the existence of a hippocampal convergence state that would predict successful subsequent memory formation, and moreover would reflect univariate activation of dopaminergic midbrain ventral tegmental area.

In the current study, we investigated relationships among motivation, midbrain VTA activation, anticipatory hippocampal states, and subsequent memory formation in an fMRI study of participants reading trivia questions and awaiting their answers. To operationalize putative convergence states, we devised a novel MVPA approach. Briefly, the activation pattern for each trial was operationalized as a point in an N-dimensional state space (with N voxels), and a centroid that represented the prototypical neural state (i.e., spatially distributed patterns) was defined using data from independent scan runs. We then calculated the typicality of each state, that is, its trial-level deviation from the prototypical state centroid. Patterns closer to the centroid show higher typicality and were thus considered to be more convergent than patterns further from the centroid. We applied convergence analysis to fMRI data acquired while participants engaged in a trivia quiz paradigm designed to elicit anticipatory states associated with either high or low motivation to learn, here curiosity. Replicating findings from prior work[18,19], we found that high motivation was associated with better subsequent recall, and with greater activation in the midbrain VTA during anticipation of answers. Using our novel analysis, we showed that higher anticipatory pattern typicality in the hippocampus, but not in the medial temporal cortex, was strongly modulated by curiosity state. Higher anticipatory typicality in the hippocampus was also predictive of better subsequent recall. Crucially, hippocampal pattern typicality was strongly associated with trial-by-trial anticipatory midbrain VTA activation, and uniquely accounted for the significant association between greater midbrain VTA activation and greater subsequent memory recall. Together, these findings support the notion of a hippocampal convergence state conducive to memory formation, and further suggest instantiating a convergence state as a candidate mechanism for neuromodulation of memory.

## Results

### Memory recall was better for high-curiosity than low-curiosity trivia

During an fMRI session, participants viewed trivia questions they had previously rated as eliciting different levels of curiosity, each followed by its answer after a variable time interval (Fig. 1a). Only

trivia questions that the participant indicated not knowing the answer to were included for the fMRI session (see "*Trivia question stimulus screening*" in the Methods section for details). In a memory test following the fMRI session, participants were presented with the trivia questions, and were required to recall the associated answer. Consistent with the expectation of enhanced learning in a motivated state, participants recalled more answers to trivia questions that had previously elicited higher levels of curiosity than those eliciting lower levels of curiosity ($t(22) = 9.32$, $p < 0.001$, $d = 1.94$, mean difference = 0.23, 95% CI = [0.18 0.28], High Curiosity: M = 0.63, SD = 0.17; Low Curiosity: M = 0.40, SD = 0.16, Fig. 1b). This finding remained robust when controlling for self-reported likelihood of knowing, and when curiosity ratings were used as continuous predictors of memory (Supplementary Results).

## Midbrain VTA activation during anticipation of answers increased after high-curiosity questions and predicted better recall

We used mixed-effects models to examine if trivia questions eliciting higher curiosity also evoked greater anticipatory activation in the mesolimbic midbrain VTA, hippocampus, and the medial temporal cortices, regions that have been associated with enhanced learning in a

motivated state (Fig. 2; refer to "*Univariate analysis - Effects of Curiosity on anticipatory activity*" in the Methods section for model specifications). In line with evidence of midbrain engagement during motivated learning, anticipatory activation in the midbrain VTA was greater following the presentation of high curiosity questions than following the presentation of low curiosity questions ($b = 0.099$, SE = 0.045, $p = 0.026$). In the medial temporal cortices, the perirhinal cortex showed a similar effect of curiosity ($b = 0.085$, SE = 0.036, $p = 0.017$), with greater anticipatory activation for high curiosity than low curiosity questions. While this trend was also observed in the parahippocampal cortex, it did not reach statistical significance ($b = 0.08$, SE = 0.042, $p = 0.056$). In contrast to the medial temporal cortices, anticipatory activity in the hippocampus was not significantly different between curiosity states ($b = 0.019$, SE = .040, $p = 0.63$).

To examine whether anticipatory activation was related to memory for subsequently presented trivia answers, we used a mixed-effects logistic regression approach with trial-level univariate activation of all ROIs included as predictors of subsequent recall (i.e., a separate regressor for each ROI). This approach allows the identification of variance that is uniquely accounted for by each of the ROIs (Fig. 3). Consistent with prior findings, VTA was a significant predictor of subsequent recall, such that greater VTA activation was associated

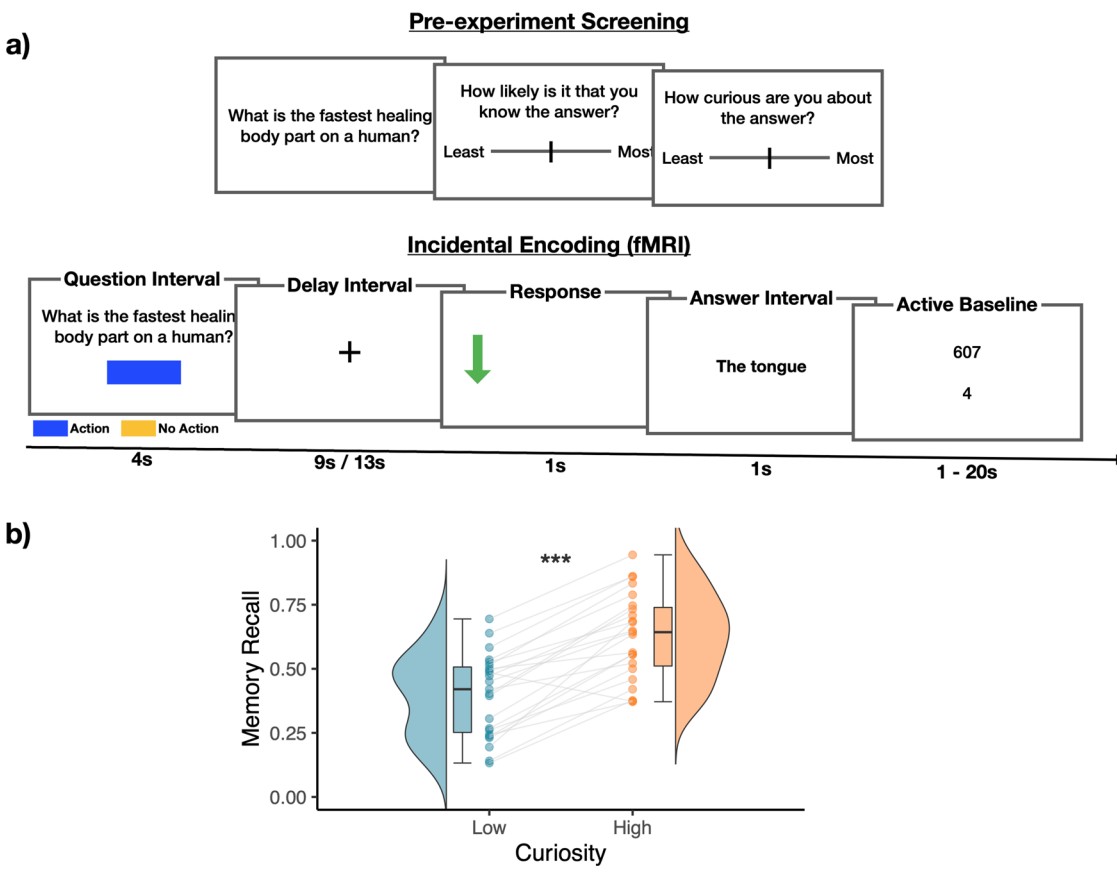

**Fig. 1 | Task schematics & Memory performance. a** Prior to fMRI scanning, participants were shown a series of trivia questions. For each, they were told to indicate the likelihood that they knew the answer, and how curious they were about it. Questions were excluded if participants indicated a likelihood of knowing the answer above 90%. The remaining questions were separated into tertiles to be used for fMRI scanning, with the 1st and 3rd tertile categorized as Low and High Curiosity questions respectively (72 questions each). During fMRI scanning, participants were shown each trivia question along with a colored rectangle that indicated the duration and action contingency of the trial. On action-contingent trials, an arrow was presented after a 9 s or 13 s delay. Participants indicated the direction of the arrow with a button press. This was followed by the presentation of the trivia answer. On non-action contingent trials, the trivia answer was presented

immediately after the delay interval. Following the scan, participants were shown each trivia question and were required to recall its associated answer. Analyses of answer Anticipation used activation evoked by each question (the Question Interval). Analyses of answer encoding used activation evoked by each question (Answer Interval, including the response on action-contingent trials). **b** Box plots for memory recall performance across each condition. Memory recall was better for high- (orange) than low-curiosity (blue) trials based on two-tailed paired $t$-test ($t(22) = 9.32$, $p < 0.001$, $d = 1.94$, mean difference = 0.23, 95% CI = [0.18 0.28]; $N = 23$ participants). The center of each box corresponds to the mean value, the upper and lower hinges correspond to the first and third quartiles, while the whiskers correspond to the largest and smallest values within 1.5 times of the interquartile range. Each dot corresponds to the recall performance of each participant. ***$p < 0.001$.

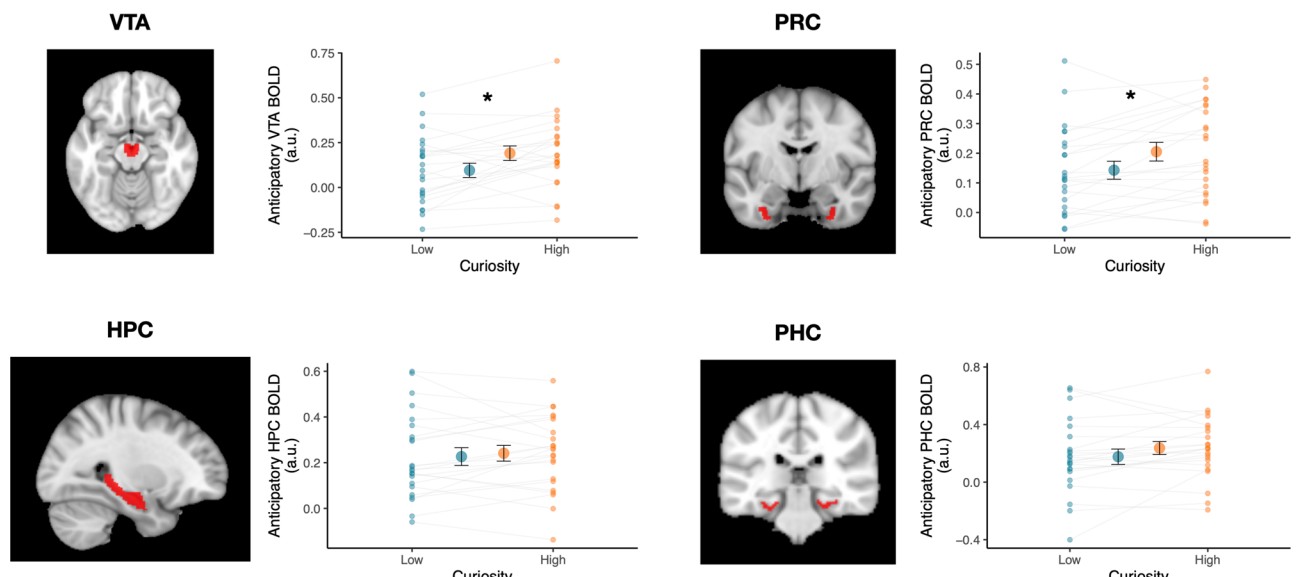

**Fig. 2 | Curiosity increased univariate activation in the midbrain VTA and perirhinal cortex during anticipation of trivia answers.** Anticipatory BOLD activation (i.e., during the Question interval preceding each trivia answer) was greater after High-Curiosity (orange) versus Low-Curiosity (blue) trivia questions in the midbrain VTA ($p = 0.026$) and perirhinal cortex (PRC) ($p = 0.017$; $N = 23$ participants). Hippocampus (HPC) and parahippocampal cortex (PHC) activation did not differentiate curiosity states. Red overlays on the brain images demarcate the ROIs (Brain template obtained from FSL and reproduced with permission). The large dots in each panel represent the group mean; small dots represent mean activation for each participant. Error bars represent the SEM. *$p < 0.05$.

with a greater likelihood of recall ($b = 0.121$, SE $= 0.049$, $p = 0.015$). Univariate activation in all other ROIs was not a significant predictor of memory outcome. To ensure that this was not simply driven by greater univariate activity for the High curiosity questions, a model comparison was performed comparing model fit between a model with and without an interaction term for the Curiosity condition. The inclusion of an interaction term for VTA activation and Curiosity did not result in a better model fit ($\chi^2 = 1.72$, $p = 0.190$), suggesting that fluctuations in anticipatory midbrain VTA activation are related to subsequent recall performance regardless of curiosity states.

### Hippocampal pattern typicality during anticipation of answers increased after high-curiosity questions and predicted better recall

Given our hypothesis that the motivation to learn would bias anticipatory hippocampal states toward a successful encoding state that followed the Anna Karenina principle, we devised an approach to characterize the *convergence of* hippocampal states based on their pattern typicality or the distance from a prototypical state centroid (Fig. 4a; refer to *"Multivariate convergence analysis"* in the Methods section for details). For this convergence analysis, we used a leave-one-run out approach where a cluster centroid was defined using data from N-1 runs. The multivariate activation pattern for each trial can be operationalized as a point in a high dimensional state space. The centroid is the point with the shortest distance (Pearson's correlation distance) to all other points in the state space. This centroid was then used as the origin to quantify the distance for trials from the left-out run. This was repeated for all runs, and each trial was assigned a value representing the distance between the activation pattern for that trial, and the independently defined centroid. In the current formulation, patterns closer to the centroid (i.e., shorter distances), thus showing greater typicality, are considered to exhibit greater convergence than patterns further from the centroid.

We first examined whether the convergence of anticipatory states in the medial temporal lobe ROIs was influenced by curiosity (Fig. 4b). Consistent with the expectation that a motivation to learn may bias hippocampal state, we observed a significant main effect of curiosity,

such that spatially distributed patterns in the hippocampus showed greater typicality during high-curiosity anticipation than during low-curiosity ($b = -0.017$, $SE = 0.007$, $p = 0.011$). This effect was seen only in the hippocampus, not in the surrounding parahippocampal ($b = -0.009$, $SE = 0.011$, $p = 0.421$) or perirhinal cortices ($b = 0.003$, $SE = 0.010$, $p = 0.788$).

To examine whether state convergence during anticipation predicted subsequent recall (Fig. 4c), we used a mixed effect logistic regression, with pattern typicality of all medial temporal lobe ROIs as predictors of recall. Pattern typicality in the hippocampus was the only significant predictor of subsequent recall. Thus, anticipatory activation patterns that were more typical were also associated with a higher likelihood of successful memory formation ($b = -0.55$, SE $= 0.21$, $p = 0.008$). Like the effects of curiosity, the relationship of anticipatory pattern typicality to memory was also specific to the hippocampus, and not seen in the surrounding medial temporal cortices (Parahippocampal cortex: $b = 0.06$, $p = 0.622$; Perirhinal cortex: $b = 0.06$, $p = 0.634$). While anticipatory univariate activation in the hippocampus was not a significant predictor of subsequent recall, we ran an additional control analysis including univariate signal in the hippocampus as a covariate to ensure that the convergence effect was not driven by differences in signal amplitude. Pattern typicality in the hippocampus remained a significant predictor after controlling for univariate activation ($b = -0.55$, SE $= 0.21$, $p = 0.008$), and after controlling for curiosity ratings ($b = -0.44$, SE $= 0.20$, $p = 0.029$). Moreover, the inclusion of curiosity as an interaction term improved the model fit ($\chi^2 = 191.28$, $p < 0.001$). Posthoc comparisons showed that the slope was more negative in the low curiosity condition than in the high curiosity condition ($b_{Difference} = -1.24$, SE $= 0.09$, $p < 0.001$), suggesting that low anticipatory hippocampal convergence may be particularly damaging to subsequent memory formation when the motivation to learn is lower.

### Trial-to-trial midbrain VTA activation predicted hippocampal typicality during anticipation of answers

Central to our primary hypothesis, univariate activation in the midbrain VTA during anticipation of answers was a significant

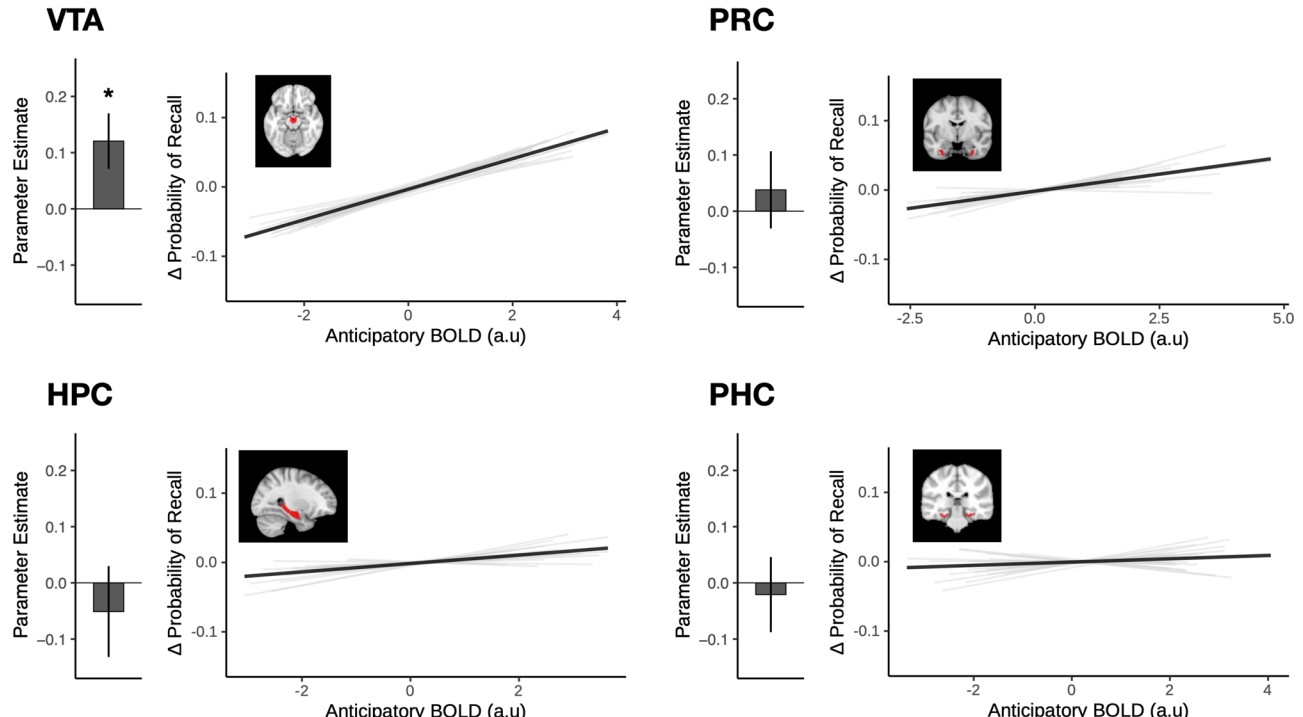

**Fig. 3 | Midbrain VTA univariate activation during anticipation of trivia answers uniquely explains subsequent recall.** Anticipatory BOLD activation (i.e., during the Question interval preceding each trivia answer) in the midbrain VTA and medial temporal lobe ROIs was used to predict memory outcome for each trial in a mixed-effects logistic regression model ($N$ = 23 participants). This method allows the identification of variance that is uniquely accounted for by each of the ROIs. Bar graphs in each panel represent the parameter estimate for each ROI in the full model. Among the ROIs, midbrain VTA activation was the only statistically significant predictor of subsequent recall of answers ($p$ = 0.015). For visualization, the estimated predicted probability of recall relative to the individual's mean probability (delta from within-subject mean, $y$-axis) is plotted against the univariate signal in each ROI (arbitrary units, a.u.; $x$-axis). Red overlays on the brain images demarcate the ROIs (Brain template obtained from FSL and reproduced with permission). Light gray lines depict the slope for each participant, while the solid black line depicts the mean slope across all participants. Error bars represent the SEM. *$p$ < 0.05.

predictor of pattern typicality in the hippocampus ($b$ = −0.059, $SE$ = 0.012, $p$ < 0.001, Fig. 5a), with greater VTA activation associated with greater hippocampal typicality (i.e., shorter distances). This result remained robust when univariate hippocampal activation was included as a covariate ($b$ = −0.049, $SE$ = 0.011, $p$ < 0.001). To ensure that this was not simply driven by differences between the curiosity states, a model comparison was performed comparing model fit between models with and without an interaction term for the Curiosity condition. The inclusion of an interaction term did not result in a better model fit ($\chi^2$ = 0.44, $p$ = 0.507), suggesting that the association between VTA activation and hippocampal typicality was not driven by differences across the curiosity states. To control for potential effects of vigilance, we also examined time-on-task effects[48,49], modeling run and trial numbers as predictors of pattern typicality. There was no significant main effect of either run ($p$ = 0.918), trial ($p$ = 0.225) or their interaction ($p$ = 0.315), suggesting that pattern typicality in the hippocampus was unlikely to be influenced by vigilance.

To examine whether pattern typicality in the hippocampus was also associated with univariate activation in other brain regions, we performed an exploratory whole-brain voxel-wise analysis correlating hippocampal typicality with each voxel's univariate activation (controlling for VTA activation). To control for spurious correlations, a null distribution was generated for each voxel using a permutation-based approach (500 iterations), and the $r$-value at the 95th percentile was subtracted from each voxel. This approach ensured that only correlation values greater than the 95th percentile of the null distribution are positive, and a one-sample $t$-test was then performed across subjects to identify regions showing a significant correlation with hippocampal convergence. At a statistical threshold of $p$ < 0.05 (FWE-corrected), we see significant clusters only in the posterior hippocampus (Peak coord: −22, −32, −2, $t$ = 10.90; Fig. 5b) and the early visual cortex (Peak Coord: 16, −86, 6, $t$ = 7.49).

### Hippocampal pattern typicality uniquely accounted for the effect of anticipatory midbrain VTA activation on memory recall

Based on our prediction that activity in the midbrain VTA influences subsequent learning by stabilizing a hippocampal convergence state, we compared mixed-effects logistic regression models that included either VTA activation, hippocampal pattern typicality, or both terms. Consistent with our hypothesis, the inclusion of both VTA activation and hippocampal typicality resulted in a better fit than a model with only VTA activation ($\chi^2$ = 5.19, $p$ = 0.023), but not a model with only hippocampal typicality ($\chi^2$ = 2.81, $p$ = 0.09). This was further supported by a mediation analysis showing a significant mediation of VTA activation by hippocampal typicality (pME = 0.276, 95% CI = [0.028 1.09], $p$ = 0.032). Together, these findings suggest that the influence of anticipatory VTA activation on subsequent recall is primarily mediated via specific effects on neural state in the hippocampus.

### Individual differences in midbrain VTA activation and hippocampal pattern typicality

Complementing the intra-individual analyses reported, we also observed a significant correlation of midbrain VTA activation and hippocampal pattern typicality across individuals, where participants with greater VTA activation showed greater typicality in the HPC ($r$ = −0.56, 95% CI = [−0.79 0.19], $p$ = 0.005). This relationship was observed for both the High curiosity ($r$ = −0.57, 95% CI = [−0.72 −0.22], $p$ = 0.004) and

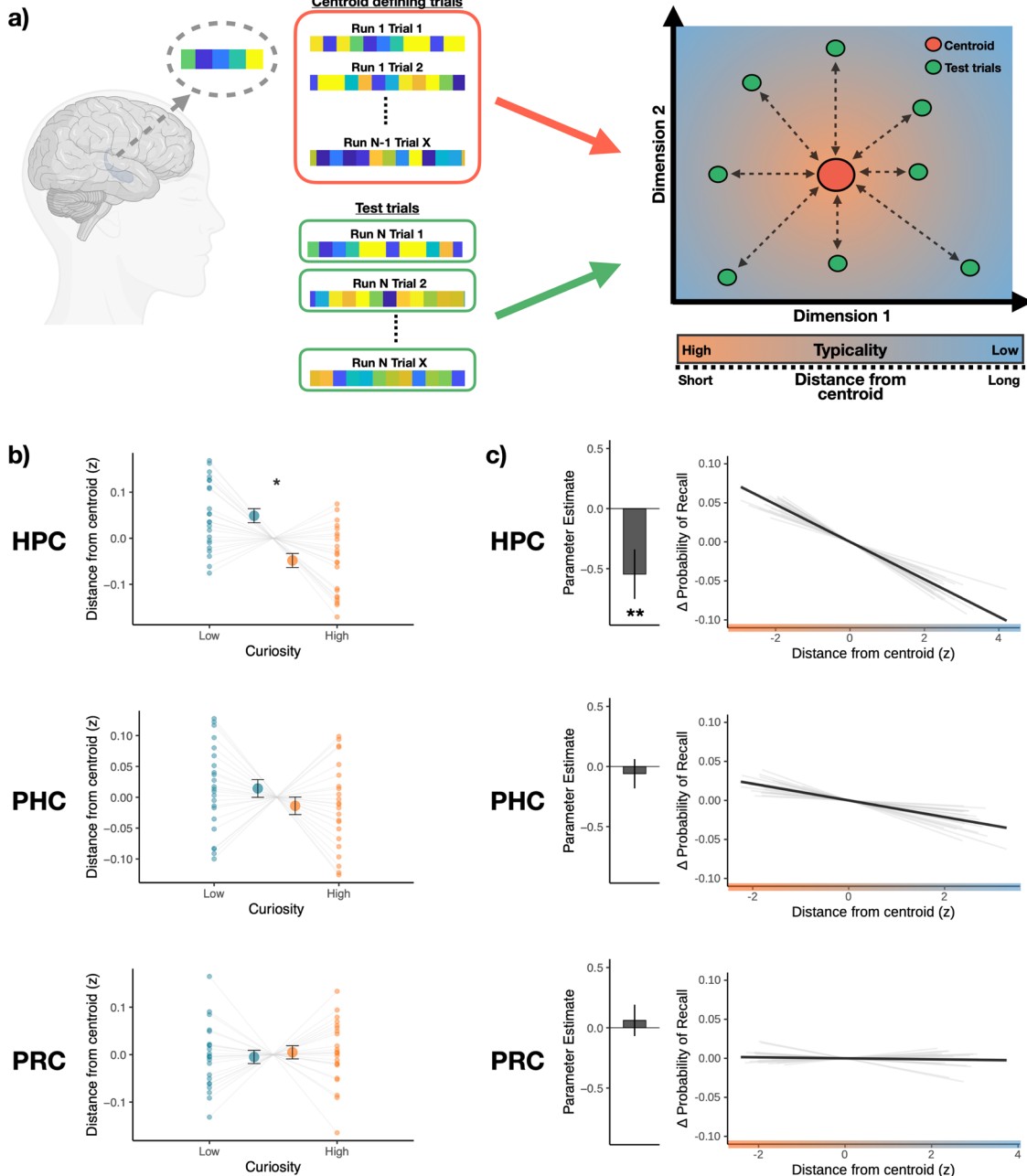

**Fig. 4 | Anticipatory pattern typicality in the hippocampus is uniquely associated with curiosity and subsequent recall. a** For the convergence state analysis, patterns of activation for each trial were extracted from each of the ROIs (Brain icon created with BioRender.com). Convergence states were operationalized as pattern typicality, or the distance between the activation pattern of each trial and an independently defined centroid representing the prototypical state). A leave-one-run-out approach was used where the cluster centroid was identified using data from N-1 runs. This centroid was then used as the origin to quantify the distance for trials from the left-out run. Patterns showing higher typicality, (i.e., shorter distance from the centroid) are considered to exhibit greater convergence than patterns more eccentric from the centroid. **b** During the anticipation of High-Curiosity (orange) trivia answers, patterns of activation exhibited higher typicality in the hippocampus (HPC, $p = 0.011$), but not the surrounding medial temporal lobe

cortices (PHC: Parahippocampal cortex; PRC: Perirhinal cortex) ($N = 23$ participants). The larger dots in each panel represents the group mean, while the smaller dots represent the mean distance for each participant. **c** We used a mixed-effects logistic regression model to predict memory outcome for each trial using the pattern typicality of the medial temporal lobe ROIs ($N = 23$ participants). Hippocampal typicality was the only significant predictor of subsequent recall ($p = 0.008$). Bar graph of each panel represents the parameter estimate of each ROI in the full model. For visualization, the estimated change in probability of recall (demeaned within subject) is plotted against the distance from centroid for each ROI. Light gray lines depict the slope for each participant, while the solid black line depicts the mean slope across all participants. Error bars represent the SEM. $*p < 0.05$, $**p < 0.01$.

Low curiosity conditions ($r = -0.44$, 95% CI = [−0.72 −0.04], $p = 0.033$). Thus, beyond intra-individual variation, individual differences in the engagement of VTA were also associated with hippocampal pattern typicality.

Additionally, we examined if curiosity-related modulation of hippocampal convergence could account for individual differences in memory performance. We did not observe a significant correlation between the individual curiosity-related hippocampal pattern typicality

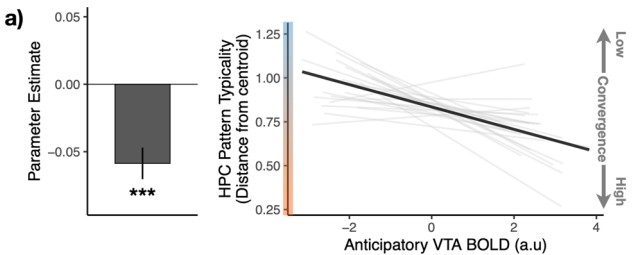

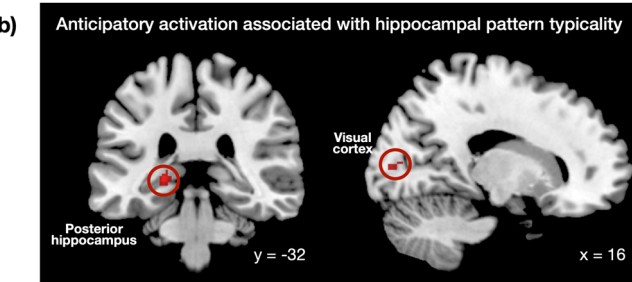

Fig. 5 | **Anticipatory midbrain VTA activation selectively modulates hippocampal pattern typicality. a** Greater activation in the midbrain VTA is associated with greater pattern typicality in the hippocampus during the anticipation of trivia answers ($p < 0.001$; $N = 23$ participants). The bar graph represents the parameter estimate for the association of VTA activation and hippocampal typicality. For visualization, the predicted distance from centroid for the hippocampus is plotted against univariate VTA BOLD activation. Light gray lines depict the slope for each participant, while the solid black line depicts the mean slope across all participants. Typicality is defined as a shorter distance from centroid. **b** Whole-brain analysis (controlling for univariate VTA activation) showed that anticipatory univariate activations in clusters that included the visual cortex and posterior hippocampus (overlaid on brain template obtained from FSL and reproduced with permission) were also positively associated with pattern typicality in the HPC (FWE $p < 0.05$). Error bars represent the SEM. ***$p < 0.001$.

(Low - High curiosity) and memory benefits ($r = -0.05$, 95% CI = [−0.45 0.37], $p = 0.831$). Instead we found a significant correlation between curiosity-related hippocampal typicality and overall memory performance (across both conditions), whereby a greater curiosity-related hippocampal typicality was associated with better memory performance overall ($r = 0.44$, 95% CI = [0.03 0.72], $p = 0.037$). An additional control analysis also showed that individual hippocampal pattern typicality was not correlated with motion (absolute RMS; $r = 0.15$, 95% CI = [−0.27, 0.53], $p = 0.475$).

### During encoding of answers, pattern typicality in the medial temporal cortices predicted subsequent recall, unrelated to curiosity

While our primary focus was on the anticipatory state following the presentation of the trivia questions, for completeness, we also conducted the same analyses on activation associated with encoding, during the presentation of trivia answers. High curiosity was associated with a trend towards greater univariate activation in the VTA ($b = 0.059$, $SE = 0.031$, $p = 0.057$). None of the medial temporal lobe ROIs showed a significant effect of curiosity during the presentation of answers (Hippocampus: $b = -0.020$, $SE = 0.028$, $p = 0.481$; Parahippocampal cortex: $b = -0.017$, $SE = 0.029$, $p = .574$; Perirhinal cortex: $b = 0.036$, $SE = 0.024$, $p = 0.127$). Across all ROIs, only activation in the perirhinal cortex was associated with a greater likelihood of recall ($b = 0.223$, $SE = 0.076$, $p = 0.003$). These findings held when activation across both intervals (Questions & Answers) were included in a single model (Supplementary Fig. 1), suggesting that univariate activation during the presentation of questions and answers each account for unique variance related to memory outcomes.

We also performed convergence analyses assessing pattern typicality during the encoding of trivia answers (Supplementary Fig. 2).

Curiosity did not predict pattern typicality during answers in any of the medial temporal lobe ROIs (Hippocampus: $b = 0.002$, $SE = 0.008$, $p = 0.750$; Parahippocampal cortex: $b = 0.006$, $SE = 0.011$, $p = 0.558$; Perirhinal cortex: $b = -0.009$, $SE = 0.010$, $p = 0.361$). Unrelated to curiosity, pattern typicality during answers in the parahippocampal ($b = -0.500$, $SE = 0.133$, $p < 0.001$) and perirhinal ($b = -0.605$, $SE = 0.139$, $p < 0.001$) cortices, but not the hippocampus ($b = 0.124$, $SE = 0.184$, $p = 0.500$), significantly predicted subsequent recall. This remained significant after controlling for univariate activation in both the parahippocampal ($b = -0.379$, $SE = 0.132$, $p = 0.004$) and perirhinal cortices ($b = -0.558$, $SE = 0.137$, $p < 0.001$).

## Discussion

The current study identifies a unique predictor of subsequent memory- anticipatory multivariate pattern typicality in the hippocampus, consistent with our proposal of a hippocampal convergence state conducive to successful memory formation. After trivia questions eliciting high curiosity, patterns of activation in the hippocampus, but not the surrounding medial temporal cortices, were biased towards greater anticipatory typicality, and anticipatory hippocampal typicality was uniquely associated with later recall of the anticipated answers. Across individuals, curiosity-related hippocampal typicality was correlated with overall recall. Most importantly, the typicality of hippocampal patterns was strongly associated with trial-by-trial variation in anticipatory univariate midbrain VTA activation, and uniquely accounted for the relationship between midbrain VTA activation and subsequent recall. These findings suggest that when higher curiosity during anticipation of answers engaged midbrain VTA activation, a multivariate pattern exhibiting high typicality (i.e., convergence state) simultaneously emerged in the hippocampus, predicting memory formation. This implied cascade points toward potential answers to long standing questions about neuromodulation and about fundamental hippocampal memory mechanisms.

Our approach leverages prior work in which fMRI multivoxel pattern analysis was used to examine representations of content during encoding. In prior studies, similarity in neural patterns across repeated occurrences of the same stimulus was associated with better memory, attributed to reinstatement of stimulus specific information[50–52]. Similar analytical approaches comparing patterns of activation across consecutive timepoints have also been used to examine temporal dynamics in the hippocampus[53]. Here, we used pattern analysis not to study stimulus representations or temporal dynamics, but rather to identify consistent engagement of states or cognitive processes associated with the anticipation of answers and successful memory formation. Our convergence analysis calculated the distance from a prototypical activation pattern that includes trials from all conditions, and patterns associated with subsequent remembering exhibited greater "typicality", while patterns associated with forgetting showed greater eccentricity. This dissociation is consistent with the expectation that successful memory formation is likely to require the convergence of multiple cognitive operations and physiological conditions, while memory failures can arise from disruption to any of the component elements, following the Anna Karenina principle. In the domain of human neuroimaging, this principle has been previously applied to examine individual differences[40]. Here, by showing that hippocampal convergence was positively associated with both activation in the midbrain VTA and with subsequent recall (as summarized in Fig. 6), we demonstrate that the Anna Karenina principle can similarly be applied to examine intra-individual variability in cognitive states to predict momentary behavior.

While our current approach is agnostic to underlying processes manifesting in a convergence state, a central question that remains is how an anticipatory convergence state in the hippocampus specifically benefits the formation of new memories. We consider two potential mechanisms. First, it has been suggested that the hippocampus can

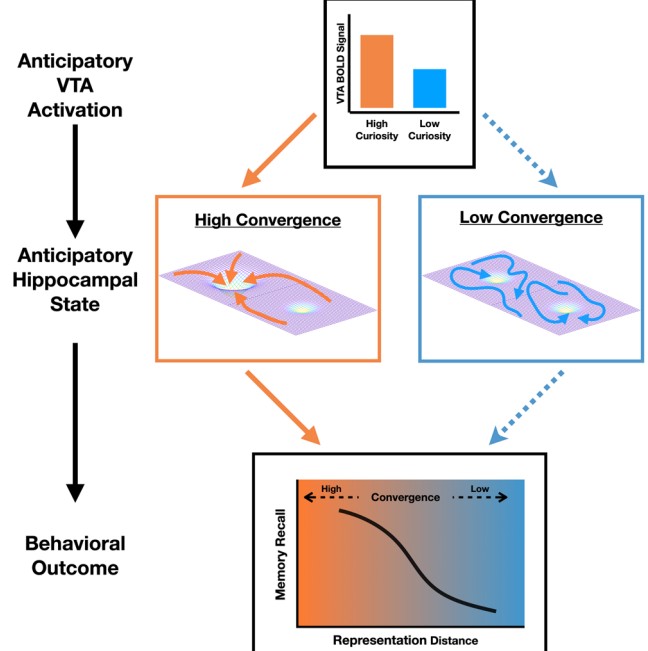

**Fig. 6 | Schematic of proposed relationships of hippocampal convergence with midbrain VTA neuromodulation.** During the anticipation of high curiosity answers, we propose that neuromodulation by the VTA promotes the consistent engagement of neural states/processes in the hippocampus that supports the formation of new memories. This can manifest as greater convergence in distributed patterns of activity. In the absence of neuromodulatory inputs from the VTA (during low curiosity), patterns of activity in the hippocampus can show greater variability. The increased convergence in hippocampal state during VTA neuromodulation may be supported by the stabilization of specific attractors and the suppression of noise-driven transitions between different possible attractor states. As hippocampal convergence and VTA activation were measured over simultaneous intervals, it is possible that hippocampal convergence supports univariate VTA activation; see text for detailed discussion.

maintain multiple spatial maps of the environment, allowing for dynamic switching between spatial frames depending on the relevant context[54]. Here, a "contextual frame" account would suggest that convergence in the hippocampus may reflect the instantiation of a "motivated context /map" that serves as the scaffold in which new memories may be embedded. This instantiation of a specific reference frame may be akin to processes underlying the "Memory palace" or the method of loci mnemonic device, where memorization is supported through the active instantiation and maintenance of a spatial reference frame that accommodates the "storage" of to-be-remembered information.

An alternative from dynamical systems is the communication subspace account[55]. It has been proposed that interareal communication across brain regions is mediated by the alignment of neural subspaces, allowing for selective propagation of neural activity[56,57]. In the memory domain, neuromodulatory activity may alter the covariance structure of neural firing within the hippocampus, thus altering the response subspace. This can facilitate the propagation of information into, and across the hippocampus, potentially suppressing noise-related activity from misaligned subspaces. This account would also build on prior proposals of the hippocampus as a mnemonic convergence zone. Indeed, our findings suggest that anticipatory midbrain activity may prime the hippocampus for optimal integration of upstream sensory information. Much in the same way that hikers may halt a conversation when attempting to locate the sound of a flowing stream, a convergent state may thus reduce noise-driven variability, supporting more efficient encoding of the incoming signal.

Related lines of work on spontaneous brain dynamics have consistently shown that variability in neural response can be reduced by exogenous stimulus[58] or by endogenous allocation of attention[42,59]. This reduction in neural variability has been associated with enhanced processing of stimulus-related information[41], and greater sensitivity in perceptual performance[42]. Computational accounts have proposed that the underlying mechanism of such reduction in neural variability may be the stabilization of specific attractors and the suppression of noise-driven transitions between different possible attractor states[60]. Though it may be tempting to attribute our current observation to similar underlying mechanisms, there are several key differences that should be noted. While the reduction in neural variability has been observed across widespread brain regions at different levels of analysis (e.g., extracellular recording[58], MRI[61], ECoG[62], EEG[63]), these are primarily observed in neocortical regions and it is uncertain if the same phenomenon would manifest in the hippocampus. In our current study, hippocampal convergence was modulated by curiosity and was predictive of subsequent recall only during anticipation (Question interval), whereas convergence in the medial temporal cortices was predictive of memory only during the presentation of information (Answer interval). These findings are consistent with an account of enhanced information processing with reduced neural variability, but also raises an intriguing possibility that convergence in the hippocampus may be more strongly modulated by internal states (curiosity in this case), whereas convergence in the medial temporal cortices may be more strongly associated with attention to external stimulus.

The idea that internal motivational states like curiosity modulate hippocampal function has now been substantiated by accumulating evidence that midbrain dopaminergic circuitry interacts with the hippocampus across multiple timescales to influence the formation of episodic memory. Neurobiological mechanisms that have been proposed to explain the effects of dopamine on memory formation in the hippocampus include meta-plastic changes, such as lowering the threshold for LTP[8,9], or "tagging" for subsequent consolidation[64]. It is unclear how, or whether, such meta-plastic changes would translate to metabolic demand, and thus changes in univariate BOLD activity during encoding. However, neuromodulation by dopamine can also alter the physiological properties of the hippocampus[1,8,9]. We reasoned that while an fMRI analytical approach cannot provide evidence about meta-plastic changes not reflected in BOLD, it is well-suited to detect modulatory influences that establish a distributed state across the hippocampus conducive to memory formation, including the convergence state we hypothesized.

It is known that the hippocampus can exhibit distinct functional states that influence how information is processed and subsequently encoded[2,34,35,38,65–69]. Computational models of episodic memory have proposed that the hippocampal memory system may alternate between functional states supporting encoding or retrieval[70–74]. While alternation between encoding and retrieval states has been suggested to occur on a rapid timescale based on theta phase, evidence from both human[75,76] and rodent[77] studies have also shown oscillatory signatures of sustained states aligned with the timescales of neuromodulatory influence. Such sustained state changes have similarly been observed using fMRI, where patterns of activation were successfully used to classify states associated with encoding or retrieval[38], and in identifying fluctuation of network states relating to encoding success[78]. In addition, behavioral studies have also shown evidence that hippocampal processing can fluctuate between a bias towards pattern separation or completion over several seconds[79,80]. As our current analytical approach focuses on anticipatory convergence states, it is unclear if our observation corresponds to any of the previously identified states, or to a theoretical "encoding-state". However, it should be noted that the current task requires not only encoding of the trivia answer, but also binding of the answer to the associated question. Although we have no separate measures of item encoding

and relational binding, we saw a dissociation consistent with their distinct roles[81–83] in item and relational memory: recall was predicted by hippocampal convergence bridging the gap between the question and answer, but by perirhinal convergence during presentation of answers alone.

While it is possible that the modulation of anticipatory hippocampal state spaces associated with midbrain VTA activation is a signature of a whole brain state that is synonymous with some usages of attention, rather than the specific VTA neuromodulation we hypothesized, we found an association between hippocampal convergence and memory only during the Question phase (i.e., preceding encoding), not during the Answer phase (i.e., during encoding). Recent human work using depth electrodes showed that pre-stimulus hippocampal spiking predicted subsequent memory but not ongoing retrieval accuracy, suggesting that hippocampal activity preceding successful encoding is unlikely to be related to attention or arousal[68]. If our findings were a result of attention, we would expect the relation between hippocampal convergence and memory to also be observed during the actual encoding of answers.

Contrasting with our findings in the hippocampus, we did observe an association of memory with perirhinal and parahippocampal convergence during encoding (i.e., the Answer phase). Cortical MTL regions are known to exhibit attention-related modulation (e.g., refs. 84,85), and in some work this has been accompanied by attention-related modulation in the hippocampus (e.g., refs. 36,37) Importantly, all of these previously reported effects of attention occurred during encoding, not during anticipation. Thus, while our cortical MTL findings could be attributable to attention, the anticipatory hippocampal convergence findings we report are not consistent with known attentional effects. In addition, hippocampal convergence remained a significant predictor of memory after the inclusion of curiosity as a covariate, suggesting that the observed relationship was unlikely to be driven by curiosity-dependent allocation of attention.

In human fMRI studies, overall increases in activation of the midbrain and the hippocampus preceding[18,19,86], during[87–90], and following encoding[91–93] have been related to better subsequent memory performance. While these prior findings are consistent with an account of dopaminergic neuromodulation, the mechanisms of such effects have remained to be specified. As discussed above, some known mechanisms of dopamine on hippocampal plasticity (metaplastic tag and capture, lowered LTP threshold) might not manifest in univariate BOLD activation. In particular, it was previously unknown how or whether spatially distributed patterns of activity in the hippocampus are modulated by midbrain activity. Our work bridges these gaps, further substantiating an account of neuromodulation of the hippocampus by midbrain activity, by showing a trial-level association between activation in the midbrain and convergence of distributed patterns in the hippocampus.

In contrast to the results of prior studies showing greater univariate activation in the hippocampus preceding successful memory formation[18,19], and despite the relationship we found between subsequent memory and pattern typicality, we found no relationship between subsequent memory and univariate signal magnitude in the hippocampus. Apart from key differences in experimental design that could have influenced anticipatory activity in prior studies, such as the use of intentional encoding[18], or the expectation of irrelevant faces[19], several analytical decisions may also have contributed to the difference in observations. In particular, our current univariate analysis was performed on the average signal across all voxels within our ROI, and was not optimized to isolate localized univariate differences. Additionally, in both of the prior studies, the mnemonic effect in the hippocampus was specific to the high motivation condition, whereas here, we showed that multivariate convergence in the hippocampus was associated with subsequent recall in both the high and low curiosity states, suggesting that distributed patterns of activity may be more

sensitive for the detection of such mnemonic effects. However, it should be noted that multivariate convergence and univariate event-related activation can co-occur, as evidenced in the medial temporal cortices during answer presentation, where both univariate activation and multivariate convergence were associated with subsequent recall. Furthermore, convergence in medial temporal cortices remained a significant predictor of recall when controlling for univariate activity, suggesting that these measures have dissociable mechanisms contributing to memory formation. We did not observe an association between anticipatory VTA and anticipatory hippocampal univariate activation. Prior findings aside, this should not be entirely surprising, because not all neuromodulatory effects that promote memory formation should be evident in changes in magnitude of BOLD activation; this premise is a main motivation for the current study. An optimal subspace account is not inconsistent with prior findings of increased anticipatory univariate activation within the hippocampus: Such activation would be expected under conditions where greater metabolic activity is required to "shift" ongoing states into the optimal subspace.

Our current contribution focuses on potential neuromodulatory influence of the hippocampus by dopaminergic nuclei in the midbrain. It should be noted that activation of the midbrain is also associated with the release of other neurotransmitters besides dopamine, and the hippocampus is also innervated by other major neuromodulatory nuclei including the basal forebrain and locus coeruleus (LC). The release of acetylcholine in the hippocampus has been shown to promote memory formation, and it has been suggested that the fluctuations in levels of acetylcholine can drive functional states in the hippocampus, with high acetylcholine promoting an encoding state and pattern separation, while low acetylcholine may promote a retrieval state and pattern completion[74,94,95]. Although our exploratory whole-brain analysis did not reveal a significant association in the basal forebrain, the primary source of cholinergic projections, this could be due to a lack of sensitivity in our current approach. It should be noted that the noradrenergic locus coeruleus has also been shown to release dopamine in the hippocampus[96,97], and recent work in mice has suggested that novelty-enhanced memory may be more strongly dependent on projections from the LC than the midbrain VTA[96]. While our current acquisition precludes a precise localization of the LC, future work should also examine if activity in the LC is similarly associated with hippocampal convergence. If hippocampal convergence is an outcome of dopaminergic modulation irrespective of its source or dynamics, then many manipulations, including novelty and prediction error, should all promote greater hippocampal convergence. We have argued previously that mesolimbic signaling to the hippocampus is more likely to be sustained than phasic[17,33]. An intriguing possibility is that hippocampal convergence is a memory mechanism specific to modulation by dopamine from mesolimbic projections.

Finally, while we evaluated a neuromodulatory hypothesis in which multivariate hippocampal convergence resulted from univariate midbrain VTA activation, it should be noted that activity in the midbrain can also be influenced, indirectly, by signaling from the hippocampus[26,98]. We have previously shown that integrating over hippocampal activation history significantly predicts midbrain VTA activation baseline drift (but not transients) in fMRI data[33]. The models used here cannot disambiguate directionality in the hippocampus-midbrain relationship. Notably, though, hippocampal convergence and midbrain activation occurred without any significant univariate hippocampal activation, so a hippocampus-to-midbrain directionality would imply that hippocampal convergence per se drove a midbrain univariate response, impacting subsequent memory. Our model comparisons and mediation analyses, moreover, showed that hippocampal convergence accounted for a greater proportion of memory variance and mediated the trial-wise relationship between VTA activation and subsequent memory. Whether convergence states manifest the effects of neuromodulation, produce it, or both, will require

further investigation in human and animal models. The chief contribution of the current work is to offer evidence for the instantiation of convergence states as a distinct hippocampal contribution to memory and as a potential route for memory modulation.

In conclusion, by using a novel systems-level characterization of hippocampal function, we identified a physiological signature consistent with an anticipatory convergence state conducive to memory formation. During high curiosity, anticipatory hippocampal convergence correlated with midbrain VTA activation and accounted for its relationship to subsequent memory, suggesting a candidate mechanism for mesolimbic memory modulation. These findings offer new tools for investigating modulation of neural states and behavior in many domains. Most fundamentally, the concept of a convergence state could potentially unite piecemeal understanding of individual mechanisms into a cohesive picture of the role of the hippocampus in memory formation.

## Methods

### Subjects
Twenty-five healthy, right-handed young adults were recruited for the study. All participants provided informed consent for our study protocol approved by the Duke University Institutional Review Board. Two participants had to be excluded (one participant fell asleep during the scan, and one did not complete the scanning session), and all remaining 23 participants were included in the analyses (10 Female; Mean age = 26.4 years, Age range = 19–35 years).

### Tasks
We selected 360 trivia questions from the stimuli used in the study by Gruber and colleagues (2014), and a pre-task screening session was used to sort trivia questions into high- and low- curiosity categories for each participant. For the pre-screening session, participants were presented with a series of trivia questions and they were required to make self-paced ratings on a continuous scale. Participants responded to the following questions: (1) "How likely is it that you know the answer?" and (2) "How curious are you about the answer?". Trivia questions were excluded if participants indicated a high likelihood of knowing the answer (>90% on the scale), and they responded until 216 trivia questions were eligible for inclusion. Included trivia questions were separated into tertiles based on curiosity ratings (72 questions each), with questions in the 1st and 3rd tertiles categorized as low and high curiosity, respectively. Twelve questions from the 2nd tertile were used as catch trials during encoding and were not included in any analysis.

Participants performed the encoding task during fMRI scanning where they were shown the trivia questions (Question Interval) and were presented with the associated answer (Answer Interval) after a variable delay interval. During the question presentation, participants were shown a single trivia question together with a colored rectangle for 4 s. The colored rectangle indicated the duration and action contingency for the upcoming trial, whereby the length of the rectangle indicated the duration of the anticipation period (9 s or 13 s), and the color indicated if participants were required to make a button press to see the trivia answer. On trials that required a button press, a green arrow appeared on the left or right side of the screen, and participants made a button press to indicate the side that the arrow was presented on. For action contingent trials, the trivia answer was shown only if the participants responded correctly or the string "XXXXX" would be presented (Participants were highly accurate and saw the trivia answer for most trials (M = 98.6%, SD = 0.4). Only trials that were correctly responded to were included for subsequent analyses). On non-action contingent trials, the trivia answer was shown at the end of the anticipation period. The manipulation of action-contingency was performed for the investigation of a separate research question that would be elaborated on in a separate communication. The trivia

answer was presented for 1 s, following which, participants performed an active baseline task where they were required to count backward from different starting numbers for a duration between 1 to 20 s. To encourage compliance with the active baseline task, catch trials occurred at random intervals, and participants were required to indicate whether their current count was above or below a given number. There were a total of 12 catch trials and trivia questions presented following the catch trials (taken from the 2nd tertile) were not included in the analysis.

Participants underwent a total of 6 scanning runs (10 min each), with 12 high curiosity trials, 12 low curiosity trials, and 2 catch trials presented within each run. Within each curiosity condition, there was an equal number of action contingent and non-action contingent trials. Condition onset and trial intervals were optimized using OptSeq2 (Dale, 1999), and the task was coded on Matlab using PsychToolBox (Brainard, 1997).

Immediately following the scan (~30 min), participants were given a surprise recall test for the trivia questions. Participants were shown all 144 (72 High and 72 Low curiosity) trivia questions in random order, and were required to type out the correct answer for each question. Participants were told not to make any guesses if they were unable to remember the correct answer.

### MRI data acquisition
MRI data were acquired on a 3 T GE Signa MRI scanner at the Duke Brain Imaging and Analysis Center. fMRI data for each participant were acquired using an echo-planar imaging (EPI) sequence (TE = 27 ms, flip angle = 77 degrees, TR = 2000 ms, voxel size = 3.75 × 3.75 mm) with 34 axial slices (slice thickness = 3.8 mm). Participants completed a total of 6 functional scan runs each consisting of 298 fMRI volumes. Cardiac and respiratory physiological data were also collected during functional scans using BioPac. Prior to the functional scans, whole-brain, inversion recovery, spoiled gradient high resolution anatomical image (voxel size = 1 mm isotropic) was collected for spatial normalization.

### fMRI preprocessing
Preprocessing of the fMRI data was performed using fMRI Expert Analysis Tool (FEAT) Version 6.00 implemented on FSL 5.0.8 (www.fmrib.ox.ac.uk/fsl). The first 6 volumes from each scan run were discarded to allow for signal stabilization. Physiological noise correction was performed using the Physiological Noise Modeling toolbox in FSL. Skull stripping was performed using BET[99], and images were realigned within-run, intensity normalized by a single multiplicative factor, spatially smoothed with a 4 mm full-width half-maximum (FWHM) kernel, and subjected to a high-pass filter (80 s). The 4 mm smoothing kernel was chosen to optimize the differentiation of midbrain and hippocampal signals[18]. Spatial normalization was performed using a two-step procedure, where mean EPI from each run was co-registered to the high-resolution anatomical image using FLIRT, which was followed by the normalization of the high-resolution anatomical image to MNI space using a nonlinear transformation with a 10 mm warp resolution implemented with FNIRT.

### Defining regions-of-interest
To examine how activity in the midbrain interacts with the medial temporal lobe (MTL), we identified regions of interest which included the midbrain VTA and regions within the MTL. The VTA was defined using a midbrain probabilistic atlas thresholded at 50% probability[100]. Three separate ROIs were defined within the medial temporal lobe (Supplementary Fig. 4), which included the hippocampus proper, perirhinal cortex and parahippocampal cortex. The hippocampus was defined using the AAL atlas, while the perirhinal and parahippocampal cortex were defined using anatomical mask from[101]. All ROIs were defined in MNI space.

## Analysis

Due to a programming error, trivia questions for one participant were not correctly selected (based on screening), and 62 trials were removed from subsequent analysis. Statistical analysis was performed using linear and logistic mixed-effects modeling using the lme4[102] and lmerTest[103] packages in R (R Core Team, 2020). Data visualization was generated using the ggplot2 package[104].

Behavioral analysis of Memory performance: Memory performance was analyzed using a paired $t$-test comparing recall rates between the high and low curiosity condition. An additional analysis was also conducted using repeated measure ANOVA with curiosity and action-contingency as factors. Post-hoc comparisons were conducted based on the contrast of estimated marginal means, and recall rate was greater for the high curiosity condition across both levels of action-contingency.

**fMRI analysis.** To capture trial-level estimates, we used the Least-squares-all approach[52,105] to estimate the betas associated with each condition at each trial. Briefly, the Question and Answer interval of each trial is modeled as a separate regressor, resulting in a total of 288 separate parameter estimates (144 trials x 2 intervals). All models also included nuisance regressors for the 6 motion parameters, and physiological (cardiac & respiratory) noise. Parameter estimates were converted to $t$-values and were then converted to $z$-values based on the inverse transform of the p-values, as implemented in FSL. Voxel values from the $z$-maps were used for both univariate and multivariate analyses.

**Univariate analysis - Effects of Curiosity on anticipatory activity.** Univariate analyses were conducted using the mean value across all voxels within each ROI during the anticipation of trivia answers (following Question presentation). Linear-mixed effects analysis was conducted for each ROI with curiosity state as fixed effect, and subjects as random effect. As previously mentioned, the manipulation of action-contingency was performed for the investigation of a separate research question, and for the current study, action-contingency was omitted from all models to increase statistical power given the limited number of trials. However, it should be noted that there was neither a main effect nor interaction of action-contingency in any of our ROIs (Supplementary Fig. 3), and the inclusion of action-contingency as a covariate did not alter any of our findings (Supplementary Results). Post-hoc comparisons were performed on the estimated marginal means using the *emmeans* package.

**Multivariate convergence analysis.** For the proposed analysis, patterns of activation in the ROIs are operationalized as points in an N-dimensional space, with N being the number of voxels in each ROI. Distance in the current analysis was measured using correlation distance (1 - Pearson's r), a distance metric commonly used in multivoxel pattern analysis. To examine the association between VTA activity and neural state in the medial temporal lobe, we devised an approach to quantify the trial-by-trial variation in neural state based on their distance from an independently defined centroid. The cluster centroid is a point with the shortest distance to all other points in high dimensional state space, and can be thought of as a prototypical state. We defined the centroid using a leave-one-run-out approach, where the cluster centroid was identified, with a k-means algorithm, using data from N-1 runs. This centroid was then used as the origin to quantify the distance for trials from the left-out run. Centroids for the analysis of the anticipatory period were defined using activation patterns from the Question interval, while analysis for the encoding of answers were defined using activation patterns from the Answer interval. This was repeated for all runs and was performed independently for each subject. The typicality for each trial was quantified based on their distance from the independently defined centroid. As the trials being measured

do not contribute to the definition of the centroid (which they are measured relative to), this approach ensures the independence of the tested trials and the centroid-defining samples. Additionally, this also ensures that the quantification of typicality is not confounded by temporal correlation (since the centroid is defined using data from a different scanning run). In the current formulation, patterns closer to the centroid (i.e., shorter distance) are considered to exhibit greater convergence than patterns further from the centroid. This operationalization is similar to measures of neural variability, whereby a larger absolute difference from the average signal amplitude is considered to reflect greater trial-to-trial variability (e.g., ref. 62). In contrast to a conventional linear classification approach, which would be suboptimal given the small number of datapoints and the imbalance between conditions in the current study (between number of Remembered and Forgotten trials), this approach also capitalizes on the expectation that successful memory formation is likely to require the confluence of multiple factors, and thus neural states conducive to memory formation should converge in state space. Linear-mixed effects analysis was conducted using the raw pattern typicality score for each ROI. To account for potential effects of outliers, key analyses were repeated after the exclusion of trials with outlying values, and all findings remained unchanged (Supplementary Results). For visualization, the measure of pattern typicality was $z$-scored across trials for each participant.

**Relating univariate activity and multivariate typicality.** To examine if multivariate pattern typicality in the hippocampus is associated with univariate activity in the midbrain VTA, a linear mixed effects model was implemented with trial-level univariate activation as a predictor of hippocampal pattern typicality. The model included subjects as random intercepts, and VTA activity as a random slope. Mediation analysis was performed using the *mediation* package.

To examine if convergence in the hippocampus is also associated with univariate activation in other brain regions, we performed an exploratory whole-brain voxel-wise analysis correlating hippocampal pattern typicality with each voxel's univariate activation (controlling for VTA activation). To control for spurious correlations, a null distribution was generated for each voxel using a permutation-based approach (500 iterations), and the $r$-value at the 95th percentile was subtracted from each voxel. This approach ensured that only correlation values greater than the 95th percentile of the null distribution are positive. A one-sample $t$-test was implemented using SPM12 (https://www.fil.ion.ucl.ac.uk/spm/), to identify regions showing a significant correlation with HPC convergence across all subjects. Significant voxels were identified using a threshold of FWE $p < 0.05$.

Relating brain activity and memory outcomes: To examine the behavioral relevance of univariate and multivariate measures of brain activity, mixed effects logistic regression was performed with trial-level brain measures (i.e., univariate activation or multivariate state convergence) of all ROIs included as predictors of subsequent recall. By including all ROIs in a single model, this approach allows the identification of variance that is uniquely accounted for by the activity of each ROIs. For all mixed-effects models, subjects were included as random intercepts, and random slopes were included if it generated a better model fit based on model comparisons evaluated using a likelihood ratio test.

### Reporting summary

Further information on research design is available in the Nature Portfolio Reporting Summary linked to this article.

## Data availability

Source data are provided with this paper and are also available at - https://github.com/JiaHou-Poh/TunedToLearn-AnticipatoryState. Source data are provided with this paper.

## Code availability

Matlab code used for convergence analysis is available at - https://github.com/JiaHou-Poh/ConvergenceState.

R code required for reproducing the results is available at - https://github.com/JiaHou-Poh/TunedToLearn-AnticipatoryState.

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

## Acknowledgements
We thank Allie Sinclair, Dr. Vishnu Murty, and Dr. Denise Cai for helpful comments on the manuscript. This work was supported by R01 MH094743 (R.A.A.), R01 MH087610 (T.E.), and the NUS Development grant (J-H.P.).

## Author contributions
Conceptualization, J.K.S., T.E., and R.A.A; methodology, J.-H.P. and R.A.A.; data collection, J.K.S; data curation, J.-H.P, J.K.S., and M.-A.T.V; code, J.-H.P; analysis, J.-H.P., M.-A.T.V., and R.A.A; data visualization, J.-H.P., and A.H.; writing-original draft, J.-H.P. and R.A.A; writing-review and editing, J.-H.P., M.-A.T.V, J.K.S., A.H., T.E., and R.A.A; funding acquisition, R.A.A. and T.E.

## Competing interests
The authors declare no competing interests.
