## [Peer Review File · Nature Communications]

Hippocampal convergence during anticipatory midbrain activation promotes subsequent memory formationREVIEWER COMMENTS

Reviewer #1 (Remarks to the Author):

In this interesting and novel fMRI study, the authors examine hippocampal memory encoding processes. Critically, the paradigm included a curiosity manipulation and the authors conclude that hippocampal 'convergence' predicts subsequent memory performance.

There is much to applaud in this manuscript. This is an exciting paper with a novel proposal about hippocampal convergence as a new candidate mechanism for efficient memory encoding. However, I think there are also some limitations of the approach and the results are at the same time somewhat preliminary.

The concept of convergence and the role of the hippocampus is not entirely new in memory research (albeit with a slightly different focus) and the authors should cover this literature: e.g. Marr 1971, McClelland, 1994, Damasio, 1989.

The key hypothesis is that 'hippocampal state spaces reflect neuromodulation'. This is a big research aim. Neuromodulation is neither directly manipulated nor directly tested but rather only indirectly manipulated (via the low vs high curiosity conditions). Midbrain activity is seen as a proxy of neuromodulation and relates to hippocampal pattern but in sum the authors just fall short in showing a convincing relationship between hippocampal state spaces and neuromodulation. The interpretation of the results needs to be adjusted accordingly - the authors simply don't show what they aimed to test.

Could increased levels of attention which might be at play during high vs low curiosity trials explain the effects? Attention needs to be convincingly controlled for.

The hippocampal pattern analysis is innovative but could higher levels of convergence simply reflect less noise in the signal? Controls for unspecific signal effects need to be provided. I think this is critical for the entire interpretation.

Reviewer #2 (Remarks to the Author):

Summary

The authors quantified the similarity of BOLD patterns in the medial temporal lobe to average patterns (called “convergence”) while participants learned the answers to trivia questions. They found that convergence in the hippocampus during the presentation of questions was positively related to self-reported curiosity about the answer, later recall of the answer, and VTA univariate responses during the question period. By contrast, convergence in the perirhinal and parahippocampal cortices during the answer presentation period was positively related to later answer recall. The researchers concluded that curiosity increases hippocampal dopaminergic innervation from the midbrain in anticipation of answers. In turn, this increased innervation triggers an anticipatory state optimal for encoding. By contrast, low dopaminergic innervation during unmotivated periods would leave the hippocampus free to engage in a variety of distinct states that are suboptimal for memory formation.

Strengths:

The paper is extremely well crafted, articulately integrating human and animal literatures to generate innovative predictions along with intriguing conjectures about potential mechanisms.

It presents the first application of the Anna Karenina principle to memory and, to my knowledge, multivariate fMRI pattern analyses. I anticipate that this exciting innovation will be of high interest to memory researchers and cognitive neuroscientists more broadly.

It provides a new way of thinking about motivated memory, linking this type of memory modulation to multivariate anticipatory brain states.

Major Analytical Comments:

(1) Why did the authors split curiosity ratings into tertials rather than retaining them as a continuous variable? None of their analyses seem to depend on this treatment, apart from the behavioral ones which would benefit from being replaced with mixed models for analytical consistency. If I understand correctly, this decision also forces the authors to discard a third of their data, in addition to offering a more limited understanding of how curiosity relates to memory and fMRI data. If the authors do not

have a good justification for this decision, I would recommend that they at least replicate their primary findings using curiosity as a continuous predictor to demonstrate that their results are not tied to this decision.

(2) The MRI modeling section lacked some key details which I would appreciate the authors including:

a. How were the question and answer intervals modeled? With a boxcar extending across the full interval or with an impulse response at the beginning of the event? Relatedly, if a boxcar was used, did the authors explore how the duration of the modeled time window relates to convergence estimates? I presume that using a larger time window would on average result in higher convergence estimates as it integrates over longer periods of time.

b. Please specify what covariates were included in each model. For example, was the anticipatory delay period included, were all other question/answer periods lumped into single regressors or stratified by curiosity, subsequent memory, etc.?

c. "Parameter estimates were converted to t-values and normalized to z-values." Was this done over space (across voxels) so that each trial had a mean of 0 or across time so that each voxel had a mean of zero? What was the justification either way? Relatedly, I find that this procedure often yields a small fraction of estimates with extreme outlying values that can be quite influential in correlation analyses. It doesn't look like the authors removed or winsorized outliers. I would strongly recommend assessing whether the data contain outliers and identifying their impact.

d. Linear-mixed effects analyses were conducted for each ROI with curiosity state as a fixed effect, and subjects as a random effect. Could the authors provide more details about these models? How were conditions coded, were only random intercepts per subject included, or were random slopes also included? If random slopes were included, how was covariance between the random effects modeled? I find it helpful when the model formula is included for clarity.

e. Related to the importance of specifying how variables were coded, did the authors guard against Simpson's Paradox and, more broadly, the contamination of within-subject analyses with between-subject variance? Common approaches include mean centering all predictors within-participant or including additional terms in the model to capture individual differences. Given that some relationships hold in across subject analyses, these steps are important to ensure that the within-subject mixed models are not simply reflecting this (separately reported) correlation across participants.

(3) An important alternative explanation of the fMRI subsequent memory results is that the observed BOLD responses reflect curiosity, not memory formation, and that curiosity's independent relationship with subsequent memory drives an apparent relationship between memory and the brain. The authors recognize this possibility, for example stating that they must "ensure that this was not simply driven by greater univariate activity for the High curiosity questions." However, they approach this confound by comparing models that contain curiosity and the interaction between curiosity and the BOLD metric to models that just contain the BOLD metric. This approach seems to be designed to ask whether curiosity

moderates the relationship between brain and memory, not whether the relationship holds when curiosity is accounted for. Could the authors instead include curiosity as a covariate to demonstrate that the BOLD metrics predict memory above and beyond their shared relationship with curiosity? This would be needed for both VTA univariate and MTL convergence analyses.

(4) The mediation analysis approach presented on pg. 16 appears better suited to answering the core question than the model comparison. I would recommend just focusing on this analysis. However, I do worry about inferring directionality here. Does VTA activation also mediate hippocampal convergence's relationship with memory? And more concerningly does memory also mediate the relationship between hippocampal convergence and VTA activity (in line with the control analyses I recommended above). I recommend that the author consider the requirements of interpreting mediation analyses in this way (e.g., temporal precedence, direct manipulation of mediator). Without a strong argument for this groundwork, I believe a more parsimonious interpretation is that all these variables share common variance and future work is required to determine why.

(5) Did the authors explore how convergence relates to motion estimates? I would anticipate that periods of sudden motion would generate high deviations from average patterns. Motion, in turn, can correlate across participants with task performance and within participants with vigilance. For these reasons, if it does correlate with convergence, I would recommend including motion estimates as covariates in convergence analyses.

(6) It appears that participants are more likely to subsequently recall than forget answers. Is this imbalance accounted for when defining a centroid for the convergence analyses? I.e., does the relationship between memory and convergence reflect the Anna Karenina principle, or might it be the result of using a centroid that on average reflects a good memory formation state?

(7) Pg. 29: "To control for spurious correlations, a null distribution was generated for each voxel using a permutation-based approach (500 iterations), and the r-value at the 95th percentile was subtracted from each voxel. This approach ensured that only correlation values greater than the 95th percentile of the null distribution are positive. A one-sample t-test was implemented using SPM12". I'm not familiar with this hybridization of non-parametric and parametric statistics. On the surface, it feels like an overcorrection on results that yield few clusters. Could the authors provide more logic, or better yet precedence for this approach?

(8) The functional scans contained relatively large voxels and MTL ROIs were using atlases: "The hippocampus was defined using the AAL atlas, while the perirhinal and parahippocampal cortex were defined using anatomical mask from." These procedures are not particularly careful ways to separate the hippocampus from surrounding MTL. The limitations of this approach are apparent in Figure 2 where the perirhinal and parahippocampal ROIs do not follow the parahippocampal gyrus. With their

small sample size, it would be quite feasible to perform individual participants segmentations using automated, manual, or hybrid procedures. Even a segmentation of the average image would be preferable.

(9) Pg. 11. “Posthoc comparisons showed that the slope was significantly greater in the low curiosity condition than in the high curiosity condition (β Difference = -1.24, SE = .09, $p < .001$), suggesting that low anticipatory hippocampal convergence may be particularly damaging to subsequent memory formation when the motivation to learn is lower.” Hard to parse because the slope of this relationship is negative. Does being greater mean that it becomes less negative in the low condition or more negative (as implied by the interpretation).

(10) The authors plan to use this data to answer multiple questions. Could they clarify whether the experiment was designed to answer the questions reported on in the current manuscript or if these are the result of exploratory analyses of a dataset designed to answer questions about the interaction of curiosity and volition? Relatedly, while the manipulation of delay interval, action-contingencies, and the cueing of these factors are not central to the current questions, it would help to know if these additional factors influence all the key variables reported here.

Major Interpretation Comments:

(1) A more thorough treatment of the behavioral data is needed to support inferences about curiosity and memory. There are many reasons why people may better remember answers to questions that generate more curiosity. One mechanism is the motivated anticipatory state that the authors focus on. Others, however, could include whether the topic is one that the participant already has some expertise in, or questions for which they already have a solid guess. Prior work that motivated this study (e.g., Gruber et al., 2014) circumvented content-related explanations by testing how well people remembered unrelated images that were presented during the anticipatory period separating questions from answers. A similar feature is not included in this study; its omission, the resulting limitations, and alternative explanations should be discussed. For example, is it possible that hippocampal convergence is higher during questions about an area of expertise, and that memories formed on the scaffolding of prior knowledge are more likely to be later remembered? The authors did collect self-reports of the likelihood of knowing the answer, discarding questions that have >90% chance of knowing ratings. This still leaves a wide range of ratings. Could the authors use this data to determine whether this proxy of prior knowledge predicts later recall and if it does, use this rating as a covariate to more precisely focus on curiosity and its relationship with memory?

(2) Throughout the paper, I found that the interpretation of findings extended beyond what the results can support. Here are just a few examples from key points in the introduction and discussion sections:

Pg. 5 “our findings suggest that neuromodulation from the VTA supports memory formation by sustaining a convergence state in the HPC that is optimal for creating memories... hippocampal convergence was strongly associated with trial-by-trial anticipatory VTA activation, and uniquely accounted for the significant association between greater VTA activation and greater subsequent memory recall.”

Pg. 17 “The present findings identify midbrain VTA neuromodulation of anticipatory hippocampal state convergence as a candidate mechanism of motivated memory.... We conclude that higher curiosity during anticipation of answers engaged VTA activation, stabilized anticipatory convergence states specifically in the hippocampus, and thus enhanced memory formation.”

I appreciate that the presented set of results are consistent with these interpretations, however, they could also be consistent with a variety of other possibilities which are given little discussion. I described my concerns about the analyses above in more detail, but to revisit a key one – many variables are correlated in this experiment but inferences about causal directions may not be warranted. The authors do show that hippocampal convergence, VTA univariate activity, curiosity, and subsequent memory are all correlated within, and sometimes across participants. But these variables may further be correlated with unmeasured or unanalyzed variables, like vigilance/attention, and as described above, prior knowledge of trivia topic and the likelihood of already knowing the answer. Additionally, the design does not allow the disambiguation of which variables drive which. VTA univariate responses and hippocampal convergence are simultaneously measured and so their correlation may reflect interactions in either direction (as the authors only briefly note toward the end of the discussions section), or correlations with a third variable. Though less obvious, as described above, it is possible that people are most curious about trivia that happens to be most memorable for other reasons. Should the authors follow my analysis recommendations, they would be better positioned to refute some of the alternative explanations for why these variables are all correlated. I would also strongly recommend drawing clear lines between the conclusions that their data directly support and those which are a more speculative interpretation of the findings that require more empirical evidence, like that provided in Figure 6.

On a related point, the authors use the term “convergence” to describe their novel measurement throughout the paper. I am pleased that they provide a parenthetical reminder in a few places that this term simply refers to the similarity of responses to the average response across trials. However, since this term does not clearly denote what it measures, I worry that some readers may be left with the impression that this metric reflects something more, for example, measuring temporal dynamics within a trial or a concrete state that shares features across participants. I recommend that key sections (abstract, final paragraph of introduction, first and final paragraph of discussion) avoid the use of this term as much as possible, replacing it with a concrete description of what was measured to avoid misunderstandings.

(3) Although the authors argue that their evidence “speak against a purely attentional account” of their findings, it is not explained what the attentional account is. What role does attention play in this study? Could the findings of better subsequent recall have been mediated by attention? The discussion would also benefit from a section dedicated to this topic.

(4) Throughout the results and discussion sections it was difficult to keep track of which phases of a trial were going into analyses. From the methods section, I inferred that only data from the question and answer intervals (not the intervening delay intervals) were reported. This decision seems at odds with much of the interpretation of “anticipatory” convergence states preparing the hippocampus to encode the answer. If this is the focus, why not analyze the delay period? Furthermore, relating the VTA question interval response to subsequent hippocampal delay convergence could perhaps better support the causal chain laid out by the authors.

Minor comments

1. I’m not sure if this study answers the question the researchers sought out in the beginning, of “what computations or operations are altered to create a state conducive to memory formation.” Posing this question might set out the wrong expectations.
2. How much time elapsed between the trivia question stimulus screening and the fMRI session?
3. In the Method section it is stated that the approach of identifying activation patterns closer to the centroid “capitalizes on the expectation that successful memory formation is likely to require the consistent convergence of multiple factors.” This explanation is quite far removed from the analysis and, thus, is not helpful for its understanding.
4. Typo on Pg 11: “Parahippocampal cortex: $\beta = -.06$ ”
5. Fig 4: Can the authors explain why every participant’s convergence score averages to 0 across high and low curiosity?
6. Pg. 10: It would help to specify the time points used to estimate hippocampal convergence within the model.
7. Pg. 11: “To examine whether state convergence during anticipation predicted subsequent recall (Figure 4C), we used a mixed effect logistic regression. State convergence in the hippocampus was the only significant predictor of subsequent recall.” It would help to specify what the other predictors were. I presumed convergence in other ROIs
8. Pg 20. “High trial-to-trial variability in initial state could increase overlap of the subsequent encoding trajectories, reducing the distinctiveness of memory traces” This is an interesting point, but the logic is hard to follow without a brief description of the cited article. Could this be expanded a little?
9. Would be helpful if the discussion provided some speculation on how dopamine would drive BOLD convergence? There is plenty of discussion of how metaplasticity mechanisms are likely not visible in BOLD, but less space devoted to how dopamine would coordinate the transition to an optimal state.

10. Pg 26: "Spatial normalization was performed using a two-step procedure using FLIRT, where mean EPI from each run was co-registered to the high-resolution anatomical image, which was followed by the normalization of the high-resolution anatomical image to MNI space using a nonlinear transformation with a 10mm warp resolution." It sounds like the authors used FNIRT not FLIRT because a nonlinear transformation is applied. Could they also clarify if the nonlinear transformation calculated for the anatomical image was then applied to the functional images as a final step?

Reviewer #3 (Remarks to the Author):

The authors used a previously established paradigm together with fMRI and a novel analysis approach to investigate the relationship between midbrain/hippocampal interaction and memory formation. As a main findings they can show that curiosity promotes memory formation, and leads to higher hippocampal convergence. Importantly, hippocampal convergence increased with midbrain activation and accounted for the association between midbrain activation and subsequent recall. As such, these findings give novel insights into how motivational states drive hippocampal neuromodulation via the dopaminergic midbrain.

The topic under investigation is timely, the methodological approach is original, and the results support the authors claims. Moreover, the manuscript is very well written. Therefore, I very much enjoyed reading the manuscript and do not have any major concerns. The only question that remains is how the convergence analysis is influenced by the size of the region of interest? Apparently, the HC is the largest of all ROIs and therefore offers more data points. Does that play a role? If so, can this be taken into account?

Reviewer #1 (Remarks to the Author):

In this interesting and novel fMRI study, the authors examine hippocampal memory encoding processes. Critically, the paradigm included a curiosity manipulation and the authors conclude that hippocampal 'convergence' predicts subsequent memory performance.

There is much to applaud in this manuscript. This is an exciting paper with a novel proposal about hippocampal convergence as a new candidate mechanism for efficient memory encoding. However, I think there are also some limitations of the approach and the results are at the same time somewhat preliminary.

The concept of convergence and the role of the hippocampus is not entirely new in memory research (albeit with a slightly different focus) and the authors should cover this literature: e.g. Marr 1971, McClelland, 1994, Damasio, 1989.

Response: We thank the reviewer for pointing out seminal works which have been key inspiration for our own. While some of the references highlighted by the reviewer mention the concept of 'convergence', there the focus is primarily on the hierarchical organization of 'convergence zones' where multiple streams of information can be coordinated and integrated across perceptual processing. While there are conceptual overlaps, our work discusses

'convergence' as a manifestation of states under particular conditions. To disambiguate and integrate these two concepts, we have added an additional clarification in our introduction -

Pg. 5 - "While building on and sharing conceptual commonalities with prior formulations of the hippocampus as a convergence zone where multiple streams of information are integrated (e.g. Marr, 1971; Damasio 1989, McClelland, 1994; Mesulam, 1998; Backus et al., 2015), we use convergence states to indicate not only an anatomical nexus, but also co-occurrence of conditions that support the encoding of information. These conditions are not per se representations of information; they may relate to processes described using words like arousal, attention, or neuromodulation. "

In addition, after further consideration of the convergence-divergence-zone framework, we consider an intriguing possibility that our observation of pre-learning convergence may reflect a state whereby the hippocampus is primed for optimal association/integration of upstream sensory information. This has now been incorporated in our discussion, and we thank the reviewer for highlighting this possibility

Pg 21 - "This account would also build on prior proposals of the hippocampus as a mnemonic convergence zone. Indeed, our findings suggest that anticipatory midbrain activity may prime the hippocampus for optimal integration of upstream sensory information."

The key hypothesis is that 'hippocampal state spaces reflect neuromodulation'. This is a big research aim. Neuromodulation is neither directly manipulated nor directly tested but rather only indirectly manipulated (via the low vs high curiosity conditions). Midbrain activity is seen as a proxy of neuromodulation and relates to hippocampal pattern but in sum the authors just fall short in showing a convincing relationship between hippocampal state spaces and neuromodulation. The interpretation of the results needs to be adjusted accordingly - the authors simply don't show what they aimed to test.

Response: Thank you for highlighting the need to edit and clarify this important passage. The key hypothesis we aimed to test was not just the first link: '*hippocampal state spaces reflect neuromodulation,*' but the entire chain, and most crucially, the novel idea of convergence as a memory-conducive state which, as a candidate mechanism, could help explain neuromodulation of memory. We have revised this passage as detailed below to make these aims clearer.

Perhaps more central to the reviewer's concern, we would not claim to test the relationship between dopamine (or other neurotransmitter release) and hippocampal state spaces, which would of course require direct manipulations. Our aims, generally, are to understand memory mechanisms, not neuromodulation per se. We view the idea of hippocampal convergence as a mechanism for efficient memory encoding, apart from its ability to be leveraged to bias memory, as the most novel and important contribution of this work.

We agree with the reviewer's caution in noting there was no direct manipulation of dopamine or any other neurotransmitter. However, we believe that our use of 'neuromodulation' to describe our behavioral manipulation of these systems is well justified by extant literature.

First, our use of “neuromodulation” has precedent in similar work examining brain states. While neuromodulatory neurotransmitters act at the microscopic level, their influence can manifest at a macroscopic level observable with non-invasive measures like fMRI. Prior work has shown that organization of functional brain networks is related to neuromodulatory receptor expression (Shine et al. 2019b), and that BOLD activation in neuromodulatory nuclei precedes brain state transitions (Munn et al. 2021) described as “*Neuromodulation of the energy landscape*”. In a study that examined Parkinson’s patients on and off dopaminergic medication, it was also observed that network topology shifted towards greater integration when participants were off medication (Shine et al. 2019a). These reports suggest that macroscopic and network consequences of neuromodulation are observable with fMRI, and make it less likely that our usage of “neuromodulation” will be overinterpreted by readers.

Second, like the aforementioned studies, and as mentioned by the reviewer, we use midbrain BOLD activation as a proxy for neuromodulatory activity. Here, our approach is based not only on known anatomical projections (e.g. Swanson 1982; Gasbarri et al. 1994; Tsetsenis et al. 2021), and the predictions of theoretical models (Lisman and Grace 2005; Lisman, Grace, and Düzel 2011), but also on an important association between midbrain BOLD and PET measures of dopamine release (Schott et al. 2008) -- a citation we unintentionally omitted in the prior submission.

In sum, our use of ‘neuromodulation’ here does not extend beyond its prior use in the literature, and we think it will be contextualized appropriately. To help ensure this, we reviewed and edited the manuscript to further emphasize the limitations of our methods with respect to neurotransmitters, potential complications for our interpretations, and the need for targeted manipulations such as optogenetics to probe mechanistic relationships.

Finally, we would clarify that we do not think that hippocampal activation patterns or state spaces solely reflect neuromodulation, and we have reviewed the paper to ensure we do not inadvertently suggest this. Rather, we assume that neuromodulation is one factor that can influence the subspace of neural activity and activation patterns.

In our view, evidence for the idea that neuromodulation plays any role in hippocampal state spaces is important to report, because while all physiological responses in intact systems reflect both information and modulation, they are often discussed as though they are purely information. This is particularly true in the hippocampus, where coding of information and its transmittal to memory are a focus. One need look no further than the discussions of reward “contexts” and goal “representations” to see this bias. We do not intend to equate hippocampal state spaces to dopamine release or even to midbrain activation; instead, we aim to contribute evidence that spatial patterns in fMRI signal (and other physiological measures) reflect not only informational representations but also neuromodulatory processes, as an important reminder for our field and others.

Pg 3 - “Such increases in midbrain BOLD have been shown to correlate with PET measures of dopamine release in target regions (Schott et al., 2008). During anticipatory midbrain activation, and greater functional connectivity between the midbrain and regions in the medial temporal lobe, including the hippocampus (HPC), predicts memory.”

Pg 4 - "Here, we were motivated by the premise that hippocampal multivoxel activation patterns may manifest, in addition to representations of information, neural states conducive to memory formation. We reasoned that if conducive state-spaces exist, their instantiation would be a candidate mechanism for memory enhancement by neuromodulation. To isolate patterns associated with neuromodulatory effects from those representing information to be encoded, we examined intervals prior to the presentation of memoranda. We hypothesized that while awaiting valuable information, i) hippocampal states would reflect the univariate activation of dopaminergic midbrain VTA, ii) that instantiation of patterns associated with memory-conducive neural states would predict subsequent memory, and iii) that memory-conducive states would account for previously reported associations between increased midbrain VTA activation and memory formation."

Pg 5 - "Thus, we hypothesized the existence of a hippocampal convergence state that would predict successful subsequent memory formation, and moreover would reflect univariate activation of dopaminergic midbrain ventral tegmental area."

Pg 6 - "Together, these findings support the notion of a hippocampal convergence state conducive to memory formation, and further suggest instantiating a convergence state as a candidate mechanism for neuromodulation of memory."

Pg 17 - "The current study identifies a novel predictor of subsequent memory, anticipatory multivariate pattern typicality in the hippocampus, consistent with our proposal of a hippocampal convergence state conducive to successful memory formation."

Pg 18 - "These findings suggest that when higher curiosity during anticipation of answers engaged midbrain VTA activation, a multivariate pattern exhibiting high typicality (i.e., convergence state) simultaneously emerged in the hippocampus, predicting memory formation. This implied cascade points toward potential answers to long standing questions about neuromodulation and about fundamental hippocampal memory mechanisms."

Pg 24 - "Our current contribution focuses on potential neuromodulatory influence of the hippocampus by dopaminergic nuclei in the midbrain."

Could increased levels of attention which might be at play during high vs low curiosity trials explain the effects? Attention needs to be convincingly controlled for.

Response: We agree that participants are likely to be more attentive to information that they are curious about. Dopaminergic signaling has also been shown to play a key role in the regulation of attention (e.g. Nieoullon 2002). Thus, we agree in principle that attention could be one factor contributing to a convergence state set up by midbrain activity. However, several pieces of evidence are hard to reconcile with attentional effects, and suggest that attention is unlikely to account for our findings. This evidence includes 1) prior literature showing that attention effects in the hippocampus and cortical MTL are evident during encoding, but not anticipation, and 2) a

new control analysis (to address the reviewer's concern) showing that hippocampal convergence remained a significant predictor of memory after controlling for curiosity.

We found an association between hippocampal convergence and memory that was only observed during the Question phase (i.e., preceding encoding), but not during the Answer phase (i.e., during encoding). Recent human work using depth electrodes showed hippocampal activity preceding successful encoding was unrelated to attention or arousal (Urgolites et al. 2020). If our findings were a result of attention, we would expect the relation between hippocampal convergence and memory to also be observed during the actual encoding of answers. Thus, our hippocampal findings are not consistent with known attentional effects.

Contrasting with our findings in the hippocampus, we did observe an association of memory with perirhinal and parahippocampal convergence during encoding (i.e., the Answer phase). Cortical MTL regions are known to exhibit attention-related modulation (e.g. O'Craven, Downing, and Kanwisher 1999; Dudukovic, Preston, and Archie 2011), and in some work this has been accompanied by attention-related modulation in the hippocampus (e.g. Aly and Turk-Browne 2015; 2016) - but again, all these previously reported effects occurred during encoding, not during anticipation.

To address the reviewer's concern about the potential for differences in curiosity driving an attentional confound, we conducted an additional analysis where we included curiosity rating as a covariate. Hippocampal convergence remained a significant predictor of memory ($\beta = -0.44$, $SE = .20$, $p = .029$, Pg. 11), suggesting that the observed relationship was unlikely to be driven by a curiosity-dependent allocation of attention. We have now further elaborated on these findings in our discussion-

Pg. 23 - "While it is possible that the modulation of anticipatory hippocampal state spaces associated with midbrain VTA activation is a signature of a whole brain state that is synonymous with some usages of attention, rather than the specific VTA neuromodulation we hypothesized, we found an association between hippocampal convergence and memory only during the Question phase (i.e., preceding encoding), not during the Answer phase (i.e., during encoding). Recent human work using depth electrodes showed hippocampal activity preceding successful encoding was unrelated to attention or arousal (Urgolites et al., 2020). If our findings were a result of attention, we would expect the relation between hippocampal convergence and memory to also be observed during the actual encoding of answers.

Contrasting with our findings in the hippocampus, we did observe an association of memory with perirhinal and parahippocampal convergence during encoding (i.e., the Answer phase). Cortical MTL regions are known to exhibit attention-related modulation (e.g. O'Craven et al., 1999; Dudukovic et al, 2011), and in some work this has been accompanied by attention-related modulation in the hippocampus (e.g. Aly & Turk-Browne, 2015, 2016) Importantly, all of these previously reported effects of attention occurred during encoding, not during anticipation. Thus, while our cortical MTL findings could be attributable to attention, the anticipatory hippocampal convergence findings we report are not consistent with known attentional effects. In addition, hippocampal convergence remained a significant predictor of memory after the inclusion of curiosity as a covariate, suggesting that the observed relationship was unlikely to be driven by curiosity-dependent allocation of attention."

The hippocampal pattern analysis is innovative but could higher levels of convergence simply reflect less noise in the signal? Controls for unspecific signal effects need to be provided. I think this is critical for the entire interpretation.

Response: This is indeed critical for our interpretation, thank you. To clarify our position on the role of 'noise'. We believe that 'noise' could relate either to noise that is intrinsic in the biological system, or to noise in our measurement or operationalization. As suggested in our discussion, we think that convergence may indeed reflect the former, but not the latter. That is, we would agree that convergence may reflect a reduction of noise in the biological system, which may support better processing or propagation of information. This option would be consistent with our neuromodulation interpretation.

On the other hand, regarding the latter option, we do not think that the convergence findings are due to differences in measurement noise between the conditions or regions, based on several key observations. First, there was no significant difference in univariate signal amplitude within the hippocampus between the different conditions (Pg. 8). Second, we included trial-by-trial univariate signal amplitude as a covariate in our convergence analysis, and our results remained robust after accounting for signal amplitude (Pg. 11). Third, convergence analysis was performed on the z-transformed t-values of each ROI, not on the raw beta-estimates; this approach would inherently downweight the contributions of noisy voxels (Misaki et al. 2010). Fourth, all our models included regressors accounting for physiological noise (cardiac and respiratory noise) and motion (6 parameters). This is now included in Pg. 29. Finally, we conducted a control analysis examining if hippocampal convergence was associated with individual difference in motion, and we did not observe a significant correlation (Pearson's $r = .15$, 95% CI = [-.27, .53], $p = .475$). Thus, we believe that the difference in convergence between conditions is unlikely to have arisen from a difference in measurement signal-to-noise ratio.

In addition, while we have tailored our acquisitions, QA, and analyses to detection of signals in midbrain and MTL regions, we might consider potential SNR differences between regions if we had negative findings in some ROIs. However, the conditional dissociations we observed indicate sufficient sensitivity to detect relationships with convergence in all the ROIs we analyzed.

We have now also included a series of control analyses recommended by Reviewer 2, and we have shown that our observed effects are unlikely to be driven by i) motion, ii) vigilance, iii) bias in memory performance, iv) delay-related effects, v) categorical transformation of curiosity rating, or vi) prior knowledge. While it is certainly possible that we may be observing the effects of some unmeasured variables, the various control analyses reduce the likelihood that our results are driven by unspecified confounds.

We hope these new controls and the additional details provided about our methods would address the reviewer's initial concerns.

Reviewer #2 (Remarks to the Author):

Summary

The authors quantified the similarity of BOLD patterns in the medial temporal lobe to average patterns (called “convergence”) while participants learned the answers to trivia questions. They found that convergence in the hippocampus during the presentation of questions was positively related to self-reported curiosity about the answer, later recall of the answer, and VTA univariate responses during the question period. By contrast, convergence in the perirhinal and parahippocampal cortices during the answer presentation period was positively related to later answer recall. The researchers concluded that curiosity increases hippocampal dopaminergic innervation from the midbrain in anticipation of answers. In turn, this increased innervation triggers an anticipatory state optimal for encoding. By contrast, low dopaminergic innervation during unmotivated periods would leave the hippocampus free to engage in a variety of distinct states that are suboptimal for memory formation.

Strengths:

The paper is extremely well crafted, articulately integrating human and animal literatures to generate innovative predictions along with intriguing conjectures about potential mechanisms.

It presents the first application of the Anna Karenina principle to memory and, to my knowledge, multivariate fMRI pattern analyses. I anticipate that this exciting innovation will be of high interest to memory researchers and cognitive neuroscientists more broadly.

It provides a new way of thinking about motivated memory, linking this type of memory modulation to multivariate anticipatory brain states.

Major Analytical Comments:

(1) Why did the authors split curiosity ratings into tertials rather than retaining them as a continuous variable? None of their analyses seem to depend on this treatment, apart from the behavioral ones which would benefit from being replaced with mixed models for analytical consistency. If I understand correctly, this decision also forces the authors to discard a third of their data, in addition to offering a more limited understanding of how curiosity relates to memory and fMRI data. If the authors do not have a good justification for this decision, I would recommend that they at least replicate their primary findings using curiosity as a continuous predictor to demonstrate that their results are not tied to this decision.

Response: Thank you for drawing our attention to this potential source of confusion. We split the stimulus set into tertiles during the initial screening, not the data. The tertile split was performed following the individualized screening session to ensure that stimuli used for the fMRI

experiment were maximally differentiated based on subjective curiosity ratings. Only trivia questions in the 1st and 3rd tertile were presented in the scans.

No scan data was discarded due to the tertile split. Given the stimulus selection procedure, there is a discontinuity in the rating data between the high and low curiosity trivia.

However, as the reviewer suggests, we also performed a mixed effect logistic regression using the curiosity rating as a continuous variable, and we showed that curiosity remained a significant predictor of memory ($b = .48$, $SE = .034$, $p < .001$). We have now included this result on Pg 6. and in the supplementary section.

(2) The MRI modeling section lacked some key details which I would appreciate the authors including:

a. How were the question and answer intervals modeled? With a boxcar expending across the full interval or with an impulse response at the beginning of the event? Relatedly, if a boxcar was used, did the authors explore how the duration of the modeled time window relates to convergence estimates? I presume that using a larger time window would on average result in higher convergence estimates as it integrates over longer periods of time.

Response: Both question and answer intervals were modeled as a boxcar across the entire duration of the event convolved with a canonical HRF. We did not explore how the duration of the modeled time window relates to convergence estimates and we believe that there might not be a simple relation between the duration modeled and convergence estimates. The reviewer brings up an interesting factor relating to the timescale of our convergence estimates. If convergence reflects a 'state' that is stable across the anticipation period, integrating over a longer time window could provide a more reliable estimate. However, this would not necessarily yield higher typicality; for example, if convergence captures the initial state of an evolving trajectory, but trajectories are diverging, integration over a longer time window could introduce greater variability.

Understanding the timescale and dynamics of convergence is an area we are invested in, and we are currently actively developing methods to further address this question. We thank the reviewer for bringing this up and we would greatly welcome suggestions on potential methodological or experimental ideas for our future work.

b. Please specify what covariates were included in each model. For example, was the anticipatory delay period included, were all other question/answer periods lumped into single regressors or stratified by curiosity, subsequent memory, etc.?

Response: Regressors of interest included only the Question and Answer interval for each trial. All models also included nuisance regressors for the catch trials, for motion (6 motion parameters), and for physiological (cardiac & respiratory) noise.

However, we would also like to highlight an error in our methods section in which we stated that the LS-S method was used for model estimation.

This should be corrected to the LS-A method, where single-trial regressors are included in the same model. As described in the work of Mumford and colleagues (2012), there is a

disadvantage to using LS-A (over LS-S), at short ISIs (~2s), yielding larger variance, which diminishes with increasing ISI. This is not an issue in our study because intervals we used are either 9 or 13s. Furthermore, our use of z-transformed t-values (rather than the betas) should also reduce the concern for greater variance, since noisy voxels should be down-weighted (Misaki et al, 2010). We apologize for the error and have corrected our manuscript.

c. "Parameter estimates were converted to t-values and normalized to z-values." Was this done over space (across voxels) so that each trial had a mean of 0 or across time so that each voxel had a mean of zero? What was the justification either way? Relatedly, I find that this procedure often yields a small fraction of estimates with extreme outlying values that can be quite influential in correlation analyses. It doesn't look like the authors removed or winsorized outliers. I would strongly recommend assessing whether the data contain outliers and identifying their impact.

Response: The z-values used were not derived from normalization across space (voxels) or time. The conversion from t- to z- is implemented as the default in FSL, and it's based on the conversion to a p-value using the cumulative distribution function of the t-distribution, which is then inverse-transformed to a z-value. Given the direct mapping from t-to-z (given a particular degrees-of-freedom), all analyses are expected to be identical if t-values were used instead of the z-values. We have now included this clarification in the methods section of our manuscript (Pg. 29).

d. Linear-mixed effects analyses were conducted for each ROI with curiosity state as a fixed effect, and subjects as a random effect. Could the authors provide more details about these models? How were conditions coded, were only random intercepts per subject included, or were random slopes also included? If random slopes were included, how was covariance between the random effects modeled? I find it helpful when the model formula is included for clarity.

Response: Curiosity state was dummy-coded with '1' corresponding to high curiosity and '0' corresponding to low curiosity. As there were only 2 levels, they were implemented as fixed effects when included within the model. All models included subjects as random intercepts, and random slopes were included when it produces a better model fit.

As suggested by the reviewer, we have now included the model specification in the supplementary methods for greater clarity (Supplementary Materials - Model Specifications). Additionally, we have also uploaded our R markdown on - <https://github.com/JiaHou-Poh/TunedToLearn-AnticipatoryState>, for a detailed walk-through of each analysis.

e. Related to the importance of specifying how variables were coded, did the authors guard against Simpson's Paradox and, more broadly, the contamination of within-subject analyses with between-subject variance? Common approaches include mean centering all predictors within-participant or including additional terms in the model to capture individual differences. Given that some relationships hold in across subject analyses, these steps are important to

ensure that the within-subject mixed models are not simply reflecting this (separately reported) correlation across participants.

Response: The reviewer highlights an important point, which is also one that our mixed-modeling approach aims to address. As highlighted by the reviewer, the relationship between univariate VTA activation and Hippocampal convergence was also present between participants. To capture the unique within-participant effects, our mixed models included both random intercepts and random slopes (as shown in Fig 4a). This ensures that our observed association is not simply driven by individual differences.

(3) An important alternative explanation of the fMRI subsequent memory results is that the observed BOLD responses reflect curiosity, not memory formation, and that curiosity's independent relationship with subsequent memory drives an apparent relationship between memory and the brain. The authors recognize this possibility, for example stating that they must "ensure that this was not simply driven by greater univariate activity for the High curiosity questions." However, they approach this confound by comparing models that contain curiosity and the interaction between curiosity and the BOLD metric to models that just contain the BOLD metric. This approach seems to be designed to ask whether curiosity moderates the relationship between brain and memory, not whether the relationship holds when curiosity is accounted for. Could the authors instead include curiosity as a covariate to demonstrate that the BOLD metrics predict memory above and beyond their shared relationship with curiosity? This would be needed for both VTA univariate and MTL convergence analyses.

Response: As pointed out by the reviewer, modeling the interaction term allowed us to examine if the influence of anticipatory VTA activity on memory differs depending on the curiosity state. As differences between curiosity states would account for substantial variation in anticipatory VTA activity, the modeling of the interaction term should enable examination of variation in VTA activity within each curiosity condition.

As suggested by the reviewer, we also conducted additional analyses with curiosity rating as a covariate, for both the univariate and convergence analyses. For the univariate analysis, the association between anticipatory VTA activity and memory trended in a similar direction ($b = .08$, $SE = .04$), but did not reach statistical significance ($p = .07$). This was unsurprising given the effects of curiosity on anticipatory VTA activity. For the convergence analysis, Hippocampal convergence remained a significant predictor of memory ($b = -.44$, $SE = .20$, $p = .029$) after the inclusion of curiosity as a covariate.

These findings suggest that while the association of anticipatory VTA activation with subsequent recall may be primarily driven by curiosity, hippocampal convergence may capture various sources of mnemonic influences beyond the effects of curiosity. We have now included this in the results section (Pg. 11).

(4) The mediation analysis approach presented on pg. 16 appears better suited to answering the core question than the model comparison. I would recommend just focussing on this analysis. However, I do worry about inferring directionality here. Does VTA activation also mediate hippocampal convergence's relationship with memory? And more concerningly does

memory also mediate the relationship between hippocampal convergence and VTA activity (in line with the control analyses I recommended above). I recommend that the author consider the requirements of interpreting mediation analyses in this way (e.g., temporal precedence, direct manipulation of mediator). Without a strong argument for this groundwork, I believe a more parsimonious interpretation is that all these variables share common variance and future work is required to determine why.

Response: We believe that the mediation analysis binds the different pieces of our results together, but it is important that we establish the individual pieces of evidence prior to setting up the mediation analysis. This is especially important given the relative novelty of the individual hypotheses, the measures, and approach we proposed, and it is thus necessary that each association is sufficiently established and tested prior to the mediation.

The reviewer also raised two primary concerns regarding 1) whether 'VTA activation also mediate hippocampal convergence's relationship with memory' and 2) if 'memory also mediate the relationship between hippocampal convergence and VTA activity'.

With regards to Point 1, we believe that this is addressed by our model comparison, whereby the inclusion of hippocampal convergence improved the model fit over a model with only VTA activation, but not the converse. These suggest that hippocampal convergence accounts for a greater proportion of variance in memory performance over that accounted for by VTA activation. Importantly, we would also like to highlight that in the current study, the manipulation of curiosity was expected to drive activity in the VTA to modulate subsequent memory formation, as observed in prior work and as postulated by accounts of motivated memory. While it is certainly possible that curiosity may drive VTA activation, and hippocampal convergence via independent mechanisms/pathways, it should be noted that our results showed that the association between VTA activation and hippocampal convergence was not driven by differences across the two curiosity states (Pg. 14).

In addition, as suggested by the reviewer, our new control analyses showed that the findings remained robust after the inclusion of curiosity as a covariate (Pg. 11).

With regards to the latter (Point 2), we believe that this can be addressed by considering the temporal precedence of the events in question. In both the VTA and the hippocampus, we are looking at neural activity that precedes the presentation of the memoranda, i.e. the trivia answer, and as such, it is unlikely that the observed relationship between VTA activity and hippocampal convergence could be mediated by subsequent memory or any property of the memorandum that correlates with memorability. This is a prime motivation for examining anticipatory states.

We agree with the reviewer that we are unable from these analyses to make a clear directional inference. We also acknowledge and discuss the possibility that VTA activation may be driven by Hippocampal convergence (Pg. 25 and caption to Fig 6.). Based on the above, and in conjunction with the control analyses suggested above by the reviewer, we believe that our current discussion posits a viable inference based on the data, while not excluding the possibility of bi-directional influences.

Pg 25 - *"The models used here cannot disambiguate directionality in the hippocampus-midbrain relationship. Notably, though, hippocampal convergence and midbrain activation occurred*

without any significant univariate hippocampal activation, so a hippocampus-to-midbrain directionality would imply that hippocampal convergence per se drove a midbrain univariate response, impacting subsequent memory. Our model comparisons and mediation analyses, moreover, showed that hippocampal convergence accounted for a greater proportion of memory variance and mediated the trial-wise relationship between VTA activation and subsequent memory. Whether convergence states manifest the effects of neuromodulation, produce it, or both, will require further investigation in human and animal models. The chief contribution of the current work is to offer evidence for the instantiation of convergence states as a novel hippocampal contribution to memory and as a potential route for memory modulation."

(5) Did the authors explore how convergence relates to motion estimates? I would anticipate that periods of sudden motion would generate high deviations from average patterns. Motion, in turn, can correlate across participants with task performance and within participants with vigilance. For these reasons, if it does correlate with convergence, I would recommend including motion estimates as covariates in convergence analyses.

Response: We did not examine how convergence would relate to motion parameters. Sudden motion would most certainly introduce large noise, reducing the reliability of parameter estimates, but it is uncertain how that would influence the measure of convergence. Intuitively, it should be as predicted by the reviewer, whereby the introduction of large noise, should result in lower convergence. We would like to mention that all first-level models included 6 motion parameters as nuisance regressors, and thus, linear effects of motion should have already been accounted for prior to the convergence analysis. We apologize for the lack of clarity and have now included this description in our revised manuscript based on earlier suggestions from the reviewer (Methods section, Pg. 29).

To address the reviewer's concern regarding motion and convergence, we also conducted a control analysis where we correlated the mean absolute motion (RMS) for each subject with hippocampal convergence, and we did not observe a significant association between motion and hippocampal convergence (Pearson's $r = .15$, 95% CI = [-.27, .53], $p = .475$). This result has now been included in Pg. 16.

To examine the potential influence of vigilance on convergence, we draw on findings from the time-on-task literature showing that vigilance should be expected to reduce across trials (e.g. Paus et al., 1997; Sun et al., 2017). We modeled run and trial numbers as predictors of convergence, and there was no significant effect of either run ($p = .918$), trial ($p = .225$) or their interaction ($p = .315$). This result has now been included in Pg. 14.

(6) It appears that participants are more likely to subsequently recall than forget answers. Is this imbalance accounted for when defining a centroid for the convergence analyses? I.e., does the relationship between memory and convergence reflect the Anna Karenina principle, or might it be the result of using a centroid that on average reflects a good memory formation state?

Response: There is considerable variability in individual memory performance, but on average, participants are actually not more likely to recall than to forget answers. A paired t-test on the

numbers of trials that are recalled and forgotten showed that it was not significantly different ($t(22) = .39, p = .69$, Recall: $M=71.1$ $SD = 22.7$; Forgotten: $M = 67.5, SD = 22.1$). Therefore, the representation of remembered and forgotten trials are relatively balanced in defining the centroid.

(7) Pg. 29: “To control for spurious correlations, a null distribution was generated for each voxel using a permutation-based approach (500 iterations), and the r-value at the 95th percentile was subtracted from each voxel. This approach ensured that only correlation values greater than the 95th percentile of the null distribution are positive. A one-sample t-test was implemented using SPM12”. I’m not familiar with this hybridization of non-parametric and parametric statistics. On the surface, it feels like an overcorrection on results that yield few clusters. Could the authors provide more logic, or better yet precedence for this approach?

Response: Corrections at both within-subject and across-subjects levels of analysis have been used in several prior studies (e.g. Schlichting, Mumford, and Preston 2015; Deuker et al. 2016). Here, the permutation test was used at the individual subject level to account for the possibility that the true distribution of r-values in the data is not normally distributed around 0, while the follow-up one-sample t-test was only used as a convenient way to summarise the group data, and to allow identification of voxels that are consistently above the 95th percentile (relative to the permuted null distribution) in our sample.

While we could also use the raw (or fisher-z transformed) r-values at the individual level and perform permutation analysis at the group level (similar to that done in various packages with beta-values), this would similarly yield a hybridization of both non-parametric and parametric approaches.

It should also be noted that while permutation within-subject involves a relatively straightforward shuffling of the mapping between trial-level univariate activation at each voxel, and hippocampal convergence, permutation at the group-level can be much more complicated in the current case. For our purpose, permutation at the group-level would likely involve the shuffling of voxels across the brain and would assume interchangeability across the whole brain. While it is possible to implement spatial constraints (e.g. shuffling within anatomical regions), this would seem to add unnecessary complications. An alternative, similar to that implemented in FSL’s *Randomise* or in *SnPM*, would be to introduce random sign flips. However, as this approach would rely on the assumption of symmetrically distributed error (Nichols & Holmes, 2002), which has been questioned in fMRI (Eklund, Nichols, and Knutsson 2016) we believe that our current approach of permuting within-subjects would better control for spurious correlation at the voxel-level.

(8) The functional scans contained relatively large voxels and MTL ROIs were using atlases: “The hippocampus was defined using the AAL atlas, while the perirhinal and parahippocampal cortex were defined using anatomical mask from.” These procedures are not particularly careful ways to separate the hippocampus from surrounding MTL. The limitations of this approach are apparent in Figure 2 where the perirhinal and parahippocampal ROIs do not follow the parahippocampal gyrus. With their small sample size, it would be quite feasible to perform

individual participants segmentations using automated, manual, or hybrid procedures. Even a segmentation of the average image would be preferable.

Response: We agree that individual participant segmentation would potentially allow for greater sensitivity in several analyses, however, we believe that here individual MTL segmentation is not central to addressing our primary research question. Precise segmentation would be needed to address questions relating to subfield specificity, or to gain sensitivity in examining fine-grained content representations that may be differentiated across the MTL (e.g., along the anterior-posterior axis). While these are important questions, our current claims do not rest on these fine distinctions. While the hippocampus was the only ROI that showed an anticipatory effect associated with both curiosity and subsequent memory formation, we had originally expected that similar effects may be observed across the cortical MTL ROIs, and were not expecting a selective hippocampal effect. We think that the selectivity observed in the current study may be driven by task demands or the specific motivational context. Further disambiguation would require examination of different motivational and task contexts, and is an area we are currently investigating.

The reviewer also expressed concerns regarding the separation of each ROI, however, we believe that this concern may be unfounded, given the distinct response profile observed across all 3 MTL ROIs. In particular, the univariate effects observed in the PHC and PRC during the Answer phase are reminiscent of typical DM effects, suggesting that the activity in our ROIs is sufficiently sensitive for detecting memory-related effects. Furthermore, if the signal from the hippocampus was indeed contaminated by the surrounding cortical MTL, this should work to reduce our observed effect (since anticipatory convergence was only observed in the hippocampus, not in the cortical MTL ROIs).

The reviewer mentioned that the perirhinal and parahippocampal ROIs do not follow the parahippocampal gyrus, but we would like to highlight that the visualization of the ROI (as depicted in Figure 2) was projected onto the high-resolution single subject Colin brain image (ch2better) that is commonly used as the template background on MRICron. The Colin brain, based on multiple scans of a single subject, is not intended for precise demarcation and may have led to the wrong impression. We apologize for the confusion and agree that this may be misleading.

In Figure 2, we have now substituted the background image to the high-resolution MNI152 template, and with the same ROIs overlaid, we can see that the Perirhinal and Parahippocampal ROIs do indeed lie along the parahippocampal gyrus. For additional confirmation, we also included the same ROI overlaid on each subject's normalized anatomical scan (Figures below). Across all subjects, the PHC and PRC are well separated from the Hippocampus and are also relatively well-aligned along the parahippocampal gyrus.

We acknowledge the value of manual segmentation, and it is in our plans to further explore this effect within the hippocampal subfields in higher resolution data, but for the purpose of the current work we do not think that manual segmentation would strengthen our claims.

MTL ROIs

(9) Pg. 11. “Posthoc comparisons showed that the slope was significantly greater in the low curiosity condition than in the high curiosity condition (β Difference = -1.24, SE = .09, $p < .001$), suggesting that low anticipatory hippocampal convergence may be particularly damaging to subsequent memory formation when the motivation to learn is lower.” Hard to parse because the slope of this relationship is negative. Does being greater mean that it becomes less negative in the low condition or more negative (as implied by the interpretation).

Response: We meant to indicate that the slope is steeper in the Low condition (i.e. more negative). We apologize for the confusion, and we have edited the statement to -

Pg 11 - “Posthoc comparisons showed that the slope was more negative in the low curiosity condition than in the high curiosity condition....”

(10) The authors plan to use this data to answer multiple questions. Could they clarify whether the experiment was designed to answer the questions reported on in the current manuscript or if these are the result of exploratory analyses of a dataset designed to answer questions about the interaction of curiosity and volition? Relatedly, while the manipulation of delay interval, action-contingencies, and the cueing of these factors are not central to the current questions, it would help to know if these additional factors influence all the key variables reported here.

Response: The task was primarily designed to examine how anticipatory activation in the mesolimbic midbrain (during a motivated state), supports an ‘encoding-state’ in memory-related circuitry. While the use of our new analytical method was not planned, its development stemmed from our planned intention to examine how trial-level variability in midbrain activation modulates patterns of activation in MTL regions.

The inclusion of action-contingency as a manipulation was intended to parallel work in rodents, where anticipatory dopaminergic activity is inevitably linked to action demands. In humans, we are afforded the opportunity to examine if such anticipatory activity in the midbrain is solely driven by preparation for action, or can similarly be evoked when preparing to learn, in the absence of any explicit action requirements.

As reported in the Method section (Pg. 29), we initially included action-contingency in our task modeling, however, as there was neither a significant main effect nor interaction in any of our ROIs, we omitted this factor to increase statistical power. We have now included these results in the Supplementary section. A separate communication looking at the interaction of brain-wide networks is currently under preparation.

As suggested by the reviewer, we conducted additional control analyses for our primary findings including the delay interval as a covariate, and all results remained robust. Curiosity remained a significant predictor of hippocampal convergence ($b = -.017$, SE = .007 $p = .01$), hippocampal convergence remained a significant predictor of subsequent memory ($b = -.541$, SE = .206, $p = .008$), and VTA remained a significant predictor of hippocampal convergence ($b = -.058$, SE = .003, $p < .001$).

Major Interpretation Comments:

(1) A more thorough treatment of the behavioral data is needed to support inferences about curiosity and memory. There are many reasons why people may better remember answers to questions that generate more curiosity. One mechanism is the motivated anticipatory state that the authors focus on. Others, however, could include whether the topic is one that the participant already has some expertise in, or questions for which they already have a solid guess. Prior work that motivated this study (e.g., Gruber et al., 2014) circumvented content-related explanations by testing how well people remembered unrelated images that were presented during the anticipatory period separating questions from answers. A similar feature is not included in this study; its omission, the resulting limitations, and alternative explanations should be discussed. For example, is it possible that hippocampal convergence is higher during questions about an area of expertise, and that memories formed on the scaffolding of prior knowledge are more likely to be later remembered? The authors did collect self-reports of the likelihood of knowing the answer, discarding questions that have >90% chance of knowing ratings. This still leaves a wide range of ratings. Could the authors use this data to determine whether this proxy of prior knowledge predicts later recall and if it does, use this rating as a covariate to more precisely focus on curiosity and its relationship with memory?

Response: The reviewer raises important points about the influence of schemas and prior knowledge, which are active areas of study by memory and curiosity researchers and also in educational psychology. Studies have indeed shown that prior knowledge can influence curiosity, but this relationship is non-linear (e.g. Wade and Kidd 2019). Given their association, it may be difficult (if not impossible) to clearly disentangle the independent influence of these factors.

In our current study, curiosity was not the chief focus, but rather a means to an end. High curiosity is primarily intended as a manipulation to reliably engage midbrain activation, and to elicit a 'learning-state' that has been implied in both Gruber et al. 2014 and Adcock et al., 2006. As suggested, however, to rule out the alternate explanations the reviewer notes, we conducted additional control analyses with the self-reported likelihood of knowing as a covariate. Curiosity remained a significant predictor of both recall ($b = .74$, $SE = .083$, $p < .001$) and hippocampal convergence ($b = -.016$, $SE = .007$, $p = .02$). Self-reported likelihood of knowing was a significant predictor of recall ($b = .55$, $SE = .05$, $p < .001$), but likelihood of knowing was not a significant predictor of hippocampal convergence ($b = .0008$, $SE = .004$, $p = .832$). We have now included this control analysis in our Result section (Pg.6) with details in the Supplementary Results.

(2) Throughout the paper, I found that the interpretation of findings extended beyond what the results can support. Here are just a few examples from key points in the introduction and discussion sections:

Pg. 5 "our findings suggest that neuromodulation from the VTA supports memory formation by sustaining a convergence state in the HPC that is optimal for creating memories... hippocampal convergence was strongly associated with trial-by-trial anticipatory VTA activation... and uniquely accounted for the significant association between greater VTA activation and greater subsequent memory recall."

Pg. 17 "The present findings identify midbrain VTA neuromodulation of anticipatory hippocampal state convergence as a candidate mechanism of motivated memory.... We conclude that higher

curiosity during anticipation of answers engaged VTA activation, stabilized anticipatory convergence states specifically in the hippocampus, and thus enhanced memory formation.”

I appreciate that the presented set of results are consistent with these interpretations, however, they could also be consistent with a variety of other possibilities which are given little discussion. I described my concerns about the analyses above in more detail, but to revisit a key one – many variables are correlated in this experiment but inferences about causal directions may not be warranted. The authors do show that hippocampal convergence, VTA univariate activity, curiosity, and subsequent memory are all correlated within, and sometimes across participants. But these variables may further be correlated with unmeasured or unanalyzed variables, like vigilance/attention, and as described above, prior knowledge of trivia topic and the likelihood of already knowing the answer. Additionally, the design does not allow the disambiguation of which variables drive which. VTA univariate responses and hippocampal convergence are simultaneously measured and so their correlation may reflect interactions in either direction (as the authors only briefly note toward the end of the discussions section), or correlations with a third variable.

Though less obvious, as described above, it is possible that people are most curious about trivia that happens to be most memorable for other reasons. Should the authors follow my analysis recommendations, they would be better positioned to refute some of the alternative explanations for why these variables are all correlated. I would also strongly recommend drawing clear lines between the conclusions that their data directly support and those which are a more speculative interpretation of the findings that require more empirical evidence, like that provided in Figure 6.

Response: We agree with the reviewer’s comments that we could have more clearly delineated our speculative interpretations, and as mentioned in an earlier response to a similar concern (Response Pg. 11), we agree and acknowledge that our current approach does not allow for an unambiguous directional inference. However, we would also like to clarify why a substantial portion of our discussion focuses on the influence of VTA on the hippocampus, rather than the converse.

With regards to potential alternative accounts, we have now included additional control analyses recommended by the reviewer, and we have shown that our observed effects are unlikely to be driven by i) motion, ii) vigilance, iii) bias in memory performance, iv) delay-related effects, v) categorical transformation of curiosity rating, or vi) prior knowledge. While it is certainly possible that we may be observing the effects of some unmeasured variables, it is also important that we consider the study in the context of existing literature.

A main goal of the study is to examine how neural circuitry involved in motivation interacts with memory-related circuits to then enhance learning. Given that the behavior of interest is memory performance, the discussion thus ultimately focuses on neural circuitry thought to underlie memory formation. More importantly, our work draws heavily on the theoretical formulation of Lisman & Grace (2005, 2011), whereby the interaction between the VTA and the hippocampus constitutes a ‘loop’, with the hippocampus both signaling to and also subsequently modulated by the VTA. While it is possible that our observation can arise from

either hippocampus signaling, VTA modulation of the hippocampus, or a combination of the two, it should also be noted that we did not observe a significant univariate effect in the HPC, an effect that should be observed with HPC-VTA signaling. While this does not entirely preclude the possibility of hippocampus signaling (to the VTA), we believe that this makes it more likely that our observation captures the latter arm of the loop (i.e. VTA -> HPC).

In addition to the theoretical foundation, we also interpret our findings in light of prior experimental work demonstrating a key role of pre-learning activity in the hippocampus. In particular, recording of hippocampal activity using depth electrodes showed that pre-stimulus spiking predicted encoding success, and this pre-stimulus spiking was similarly unrelated to general arousal or attention (Urgolites et al., 2020). Our current work parallels these findings, and further demonstrates that pre-learning hippocampal states are related to activity in neuromodulatory nuclei. We agree that the statements highlighted by the reviewer included directional inferences that may not be sufficiently substantiated by our findings and we have amended the statements to more accurately reflect the actual observations -

Pg. 6 - "Together, these findings support the notion of a hippocampal convergence state conducive to memory formation, and further suggest instantiating a convergence state as a candidate mechanism for neuromodulation of memory."

Pg. 18 - "These findings suggest that when higher curiosity during anticipation of answers engaged midbrain VTA activation, a multivariate pattern exhibiting high typicality (i.e., convergence state) simultaneously emerged in the hippocampus, predicting memory formation. This implied cascade points toward potential answers to long standing questions about neuromodulation and about fundamental hippocampal memory mechanisms."

We have also now included discussion of our ability to infer directionality beginning on Pg 25 -

"Finally, while we evaluated a neuromodulatory hypothesis in which multivariate hippocampal convergence resulted from univariate midbrain VTA activation, it should be noted that activity in the midbrain can also be influenced, indirectly, by signalling from the hippocampus (Floresco, Todd, and Grace 2001; Lisman and Grace 2005). We have previously shown that integrating over hippocampal activation history significantly predicts midbrain VTA activation baseline drift (but not transients) in fMRI data (Murty, Ballard, and Adcock 2016). The models used here cannot disambiguate directionality in the hippocampus-midbrain relationship. Notably, though, hippocampal convergence and midbrain activation occurred without any significant univariate hippocampal activation, so a hippocampus-to-midbrain directionality would imply that hippocampal convergence per se drove a midbrain univariate response, impacting subsequent memory. Our model comparisons and mediation analyses, moreover, showed that hippocampal convergence accounted for a greater proportion of memory variance and mediated the trial-wise relationship between VTA activation and subsequent memory. Whether convergence states manifest the effects of neuromodulation, produce it, or both, will require further investigation in human and animal models. The chief contribution of the current work is to offer evidence for the instantiation of convergence states as a novel hippocampal contribution to memory and as a potential route for memory modulation."

On a related point, the authors use the term “convergence” to describe their novel measurement throughout the paper. I am pleased that they provide a parenthetical reminder in a few places that this term simply refers to the similarity of responses to the average response across trials. However, since this term does not clearly denote what it measures, I worry that some readers may be left with the impression that this metric reflects something more, for example, measuring temporal dynamics within a trial or a concrete state that shares features across participants. I recommend that key sections (abstract, final paragraph of introduction, first and final paragraph of discussion) avoid the use of this term as much as possible, replacing it with a concrete description of what was measured to avoid misunderstandings.

Response: We thank the reviewer for the suggestion. We carefully considered the labeling of our measure to avoid potential interference with existing constructs that are used in the field (e.g. Pattern similarity (Xue et al. 2010); Centrality (van den Heuvel and Sporns 2013); Typicality (Iordan et al. 2016)). We worried about *Typicality* because of its use in other contexts (E.g. category learning). However, we agree that while *convergence* accurately conveys the rationale for the measure, *typicality* is clearer for the measure itself. We have edited to situate these terms in appropriate contexts, with *typicality* to describe patterns and *convergence* to refer to the states we aim to detect. Below we highlight several key changes.

Abstract -

“To operationalize convergence of neural states, we quantified the typicality of multivoxel patterns in the medial temporal lobes during anticipation and encoding of trivia answers.”

Introduction-

Pg. 5 - “In the current study, we investigated relationships among motivation, midbrain VTA activation, anticipatory hippocampal states, and subsequent memory formation in an fMRI study of participants reading trivia questions and awaiting their answers.”

Pg 5 - “Briefly, the activation pattern for each trial was operationalized as a point in an N-dimensional state space (with N voxels), and a centroid that represented the prototypical neural state (i.e., spatially distributed patterns) was defined using data from independent scan runs. We then calculated the typicality of each state, that is, its trial-level deviation from the prototypical state centroid. Patterns closer to the centroid show higher typicality and were thus considered to be more convergent than patterns further from the centroid.”

Pg 6 - “... we showed that higher anticipatory pattern typicality in the hippocampus, but not in the medial temporal cortex, was strongly modulated by curiosity state. Higher anticipatory typicality in the hippocampus was also predictive of better subsequent recall. Crucially, hippocampal pattern typicality was strongly associated with trial-by-trial anticipatory midbrain VTA activation, and uniquely accounted for the significant association between greater midbrain VTA activation and greater subsequent memory recall. “

Discussion-

Pg. 17 - "The current study identifies a novel predictor of subsequent memory, anticipatory multivariate pattern typicality in the hippocampus, consistent with our proposal of a hippocampal convergence state conducive to successful memory formation. After trivia questions eliciting high curiosity, patterns of activation in the hippocampus, but not the surrounding medial temporal cortices, were biased towards greater anticipatory typicality, and anticipatory hippocampal typicality was uniquely associated with later recall of the anticipated answers."

Pg. 17 - "These findings suggest that when higher curiosity during anticipation of answers engaged midbrain VTA activation, a multivariate pattern exhibiting high typicality (i.e., convergence state) simultaneously emerged in the hippocampus, predicting memory formation. This implied cascade points toward potential answers to long standing questions about neuromodulation and about fundamental hippocampal memory mechanisms."

Pg. 18 - " Our convergence analysis calculated the distance from a prototypical activation pattern that includes trials from all conditions, and patterns associated with subsequent remembering exhibited greater 'typicality', while patterns associated with forgetting showed greater eccentricity."

(3) Although the authors argue that their evidence "speak against a purely attentional account" of their findings, it is not explained what the attentional account is. What role does attention play in this study? Could the findings of better subsequent recall have been mediated by attention? The discussion would also benefit from a section dedicated to this topic.

Response: The attentional account assumes that higher curiosity items would elicit greater attention to the answer, which could explain better subsequent recall. We have now added a section discussing this alternative explanation and our arguments against it, as highlighted in our response to Reviewer 1 as follows -

We agree that participants are likely to be more attentive to information that they are curious about. Dopaminergic signaling has also been shown to play a key role in the regulation of attention (e.g. Nieoullon 2002). Thus, we agree in principle that attention could be one factor contributing to a convergence state set up by midbrain activity. However, several pieces of evidence are hard to reconcile with attentional effects, and suggest that attention is unlikely to account for our findings. This evidence includes 1) prior literature showing that attention effects in the hippocampus and cortical MTL are evident during encoding, but not anticipation, and 2) a new control analysis (to address the reviewer's concern) showing that hippocampal convergence remained a significant predictor of memory after controlling for curiosity.

We found an association between hippocampal convergence and memory that was only observed during the Question phase (i.e., preceding encoding), but not during the Answer phase (i.e., during encoding). Recent human work using depth electrodes showed hippocampal activity preceding successful encoding was unrelated to attention or arousal (Urgolites et al., 2020). If our findings were a result of attention, we would expect the relation between hippocampal convergence and memory to also be observed during the actual encoding of answers. Thus, our hippocampal findings are not consistent with known attentional effects.

Contrasting with our findings in the hippocampus, we did observe an association of memory with perirhinal and parahippocampal convergence during encoding (i.e., the Answer

phase). Cortical MTL regions are known to exhibit attention-related modulation (e.g. O'Craven et al., 1999; Dudukovic et al, 2011), and in some work this has been accompanied by attention-related modulation in the hippocampus (e.g. Aly & Turk-Browne, 2015, 2016) - but again, all these previously reported effects occurred during encoding, not during anticipation.

To address the reviewer's concern about the potential for differences in curiosity driving an attentional confound, we conducted an additional analysis where we included curiosity rating as a covariate. Hippocampal convergence remained a significant predictor of memory ($\beta = -0.44$, $SE = .20$, $p = .029$, Pg. 11), suggesting that the observed relationship was unlikely to be driven by a curiosity-dependent allocation of attention. We have now further elaborated on these findings in our discussion-

Pg. 22 - "While it is possible that the modulation of anticipatory hippocampal state spaces associated with midbrain VTA activation is a signature of a whole brain state that is synonymous with some usages of attention, rather than the specific VTA neuromodulation we hypothesized, we found an association between hippocampal convergence and memory only during the Question phase (i.e., preceding encoding), not during the Answer phase (i.e., during encoding). Recent human work using depth electrodes showed hippocampal activity preceding successful encoding was unrelated to attention or arousal (Urgolites et al., 2020). If our findings were a result of attention, we would expect the relation between hippocampal convergence and memory to also be observed during the actual encoding of answers.

Contrasting with our findings in the hippocampus, we did observe an association of memory with perirhinal and parahippocampal convergence during encoding (i.e., the Answer phase). Cortical MTL regions are known to exhibit attention-related modulation (e.g. O'Craven et al., 1999; Dudukovic et al, 2011), and in some work this has been accompanied by attention-related modulation in the hippocampus (e.g. Aly & Turk-Browne, 2015, 2016) Importantly, all of these previously reported effects of attention occurred during encoding, not during anticipation. Thus, while our cortical MTL findings could be attributable to attention, the anticipatory hippocampal convergence findings we report are not consistent with known attentional effects. In addition, hippocampal convergence remained a significant predictor of memory after the inclusion of curiosity as a covariate, suggesting that the observed relationship was unlikely to be driven by curiosity-dependent allocation of attention."

(4) Throughout the results and discussion sections it was difficult to keep track of which phases of a trial were going into analyses. From the methods section, I inferred that only data from the question and answer intervals (not the intervening delay intervals) were reported. This decision seems at odds with much of the interpretation of "anticipatory" convergence states preparing the hippocampus to encode the answer. If this is the focus, why not analyze the delay period? Furthermore, relating the VTA question interval response to subsequent hippocampal delay convergence could perhaps better support the causal chain laid out by the authors.

Response: The primary analyses were focused on the activity evoked during the question phase, and we have edited the manuscript to make this explicit. Activity related to the anticipation of trivia answers should be evoked during question presentation. Modeling the delay period separate from this response is a complex problem with unclear incremental benefit. The

modeling of the Question and Answer interval would necessitate that the regressor for the delay period would be collinear with the regressors for the Question and Answer interval. Thus we chose to focus on the question interval. This approach is similar to that of our earlier work (Adcock et al. 2006).

While there are alternative approaches that may provide a viable means of examining delay interval activity, such as looking at background connectivity after accounting for task-evoked activity (e.g. Al-Aidroos, Said, and Turk-Browne 2012), most such alternatives would not be ideal for the same trial-level analysis that is the focus of the current work.

Minor comments

1. I'm not sure if this study answers the question the researchers sought out in the beginning, of "what computations or operations are altered to create a state conducive to memory formation." Posing this question might set out the wrong expectations.

Response: We agree that this might set up the wrong expectations and have now modified the statement as such -

Pg. 3 - "However, they do not answer the question of how activation in neuromodulatory nuclei might create hippocampal states conducive for memory formation."

2. How much time elapsed between the trivia question stimulus screening and the fMRI session?

Response: The fMRI session was conducted immediately following the stimulus screening session. We do not have a precise record of the actual time elapsed, but between the conclusion of the screening and the start of the scan, the elapsed time should be approximately 30mins. We have now included this information in the methods section (Pg. 27).

3. In the Method section it is stated that the approach of identifying activation patterns closer to the centroid "capitalizes on the expectation that successful memory formation is likely to require the consistent convergence of multiple factors." This explanation is quite far removed from the analysis and, thus, is not helpful for its understanding.

Response: The premise that convergence of multiple factors is necessary is fundamental to the Anna Karenina principles. The formulation of 'convergence' in the fMRI analysis is based on the notion that failure in different factors would lead to entropy that is equally likely across all dimensions - thus making the 'centroid' a point of convergence. To clarify, we have edited the statement to -

Pg. 30 - "...this approach also capitalizes on the expectation that successful memory formation is likely to require the confluence of multiple factors, and thus neural states conducive to memory formation should converge in state space"

4. Typo on Pg 11: “Parahippocampal cortex: $\beta = -.06$ ”

Response: The typo has been corrected.

5. Fig 4: Can the authors explain why every participant’s convergence score averages to 0 across high and low curiosity?

Response: For the purpose of visualization, we z-scored the Convergence metric across trials for each participant, and the actual scores used for the analysis do not average to 0 across high and low curiosity. We apologize for the confusion and have edited our Methods section (Pg. 31) to reflect that. We have included the non-normalized figures below for reference.

Pg. 31 - “Linear-mixed effects analysis was conducted using the raw typicality score for each ROI. For visualization, the measure of pattern typicality was z-scored across trials for each participant.”

6. Pg. 10: It would help to specify the time points used to estimate hippocampal convergence within the model.

Response: While our primary analysis focuses on anticipatory activity following Question presentation, the description on Pg 10 serves to provide a general overview of how convergence is estimated and is not specific to either the Question or the Answer period.

7. Pg. 11: “To examine whether state convergence during anticipation predicted subsequent recall (Figure 4C), we used a mixed effect logistic regression. State convergence in the hippocampus was the only significant predictor of subsequent recall.” It would help to specify what the other predictors were. I presumed convergence in other ROIs

Response: The other predictors included convergence in the other ROIs as the reviewer rightly pointed out. We apologize for the ambiguity and have edited the statement to -

Pg. 11 - “To examine whether state convergence during anticipation predicted subsequent recall (Figure 4C), we used a mixed effect logistic regression, with pattern typicality of all medial temporal lobe ROIs as predictors of recall.”

8. Pg 20. “High trial-to-trial variability in initial state could increase overlap of the subsequent encoding trajectories, reducing the distinctiveness of memory traces” This is an interesting point, but the logic is hard to follow without a brief description of the cited article. Could this be expanded a little?

Response: This account builds off the premise that different memory representations follow distinct neural trajectories, and that interference can arise due to overlaps in these trajectories. Here, we meant to indicate that variation in the initial state could reflect noise in the system that can further propagate as fluctuations in subsequent trajectory. This is similar to that depicted in Figure 1 of Zylberberg et al. 2016, whereby fluctuations spread around the ‘mean’ trajectories, increasing the overlap in otherwise distinct trajectories.

However, upon careful consideration of the reviewer’s following comment (Point 9), we believe that rather than the optimal subspace account, the communication subspace account may better account for our observation. As such, we have amended our discussion (Pg. 22) as such -

Pg. 21 - “An alternative from dynamical systems is the communication subspace account (Semedo et al., 2020). It has been proposed that interareal communication across brain regions is mediated by the alignment of neural subspaces, allowing for selective propagation of neural activity (Semedo et al., 2019; Yoo & Hayden 2020). In the memory domain, neuromodulatory activity may alter the covariance structure of neural firing within the hippocampus, thus altering the response subspace. This can facilitate the propagation of information into, and across the hippocampus, potentially suppressing noise-related activity from misaligned subspaces”.

9. Would be helpful if the discussion provided some speculation on how dopamine would drive BOLD convergence? There is plenty of discussion of how metaplasticity mechanisms are likely

not visible in BOLD, but less space devoted to how dopamine would coordinate the transition to an optimal state.

Response: This is something that we have thought deeply about and we thank the reviewer for this question. In addition to the metaplasticity mechanisms that we have mentioned, neuromodulators can also change the covariance structure of neural firing (e.g. Thiele and Bellgrove 2018). Computationally, this would manifest as an alteration in the subspace/manifold. One potential mechanism in which dopamine might drive convergence is by increasing the covariance of neural firing, thus ‘collapsing’ the available subspace to a low-dimensional manifold. By constraining the available subspace, the impact of noise can be greatly reduced, as activity ‘misaligned’ with the relevant manifold is not further propagated across the processing stream (e.g. Smedo et al. 2019; Yoo and Hayden 2020).

While we have initially chosen to omit this speculative account in our original manuscript, after careful consideration prompted by the reviewer’s comment, we decided that the communication subspace account may provide greater clarity towards understanding the underlying biological and computational mechanisms. As such, we have amended our discussion as highlighted in our response to the previous comment (Pg. 21).

10. Pg 26: “Spatial normalization was performed using a two-step procedure using FLIRT, where mean EPI from each run was co-registered to the high-resolution anatomical image, which was followed by the normalization of the high-resolution anatomical image to MNI space using a nonlinear transformation with a 10mm warp resolution.” It sounds like the authors used FNIRT not FLIRT because a nonlinear transformation is applied. Could they also clarify if the nonlinear transformation calculated for the anatomical image was then applied to the functional images as a final step?

Response: Both FLIRT and FNIRT were used for the linear- (EPI to anatomical) and nonlinear (anatomical to MNI) components respectively. The transformation matrices from both steps are combined and applied to the functional images as implemented in FSL’s FEAT. We thank the reviewer for spotting the omission and have corrected this in our revised manuscript (Pg. 28).

Reviewer #3 (Remarks to the Author):

The authors used a previously established paradigm together with fMRI and a novel analysis approach to investigate the relationship between midbrain/hippocampal interaction and memory formation. As a main finding they can show that curiosity promotes memory formation, and leads to higher hippocampal convergence. Importantly, hippocampal convergence increased with midbrain activation and accounted for the association between midbrain activation and subsequent recall. As such, these findings give novel insights into how motivational states drive hippocampal neuromodulation via the dopaminergic midbrain. The topic under investigation is timely, the methodological approach is original, and the results support the authors' claims. Moreover, the manuscript is very well written. Therefore, I very much enjoyed reading the manuscript and do not have any major concerns. The only question that remains is how the convergence analysis is influenced by the size of the region of interest? Apparently, the HC is the largest of all ROIs and therefore offers more data points. Does that play a role? If so, can this be taken into account?

Response: We do not have a clear prediction on how the number of voxels would influence the measure of convergence. The hippocampus is indeed the largest of the ROIs, and theoretically, a greater number of voxels should increase the dimensionality of the state space, rendering convergence less likely (in larger state space). However, this relationship might not be linear, given the covariance of voxels/dimensions.

Here, we offer two reasons why our results are unlikely to be driven by differences in voxel numbers. First, the comparisons of convergence were conducted between conditions within the same ROI (e.g., High vs Low curiosity), so that the number of voxels is constant. Second, we observed greater convergence during the Question phase predicting memory in the larger hippocampal ROI, but greater convergence during the Answer phase predicting memory in the smaller parahippocampal cortex and perirhinal cortex ROIs, rather than in the hippocampus. This double dissociation by condition suggests that the difference in size between ROIs does not impact the sensitivity for detecting convergence effects.

The reviewer's comment also highlights an interesting relation regarding dimensionality and convergence. Given the low dimensional manifold commonly observed in neural data, it would be interesting to examine if the convergence state reflects a manifestation of such a low dimensional manifold (i.e. expression of 'state' in a low dimension, thus leading to shorter distances). This is an area of interest that we are currently working on, and we thank the reviewer for the keen observation.

References

- Adcock, R. Alison, Arul Thangavel, Susan Whitfield-Gabrieli, Brian Knutson, and John D. E. Gabrieli. 2006. "Reward-Motivated Learning: Mesolimbic Activation Precedes Memory Formation." *Neuron* 50 (3): 507–17.
- Al-Aidroos, N., C. P. Said, and N. B. Turk-Browne. 2012. "Top-down Attention Switches Coupling between Low-Level and High-Level Areas of Human Visual Cortex." *Proceedings of the National Academy of Sciences* 109 (36): 14675–80.
- Aly, Mariam, and Nicholas B. Turk-Browne. 2016. "Attention Promotes Episodic Encoding by Stabilizing Hippocampal Representations." *Proceedings of the National Academy of Sciences*, January, 201518931–201518915.
- Aly, M., and N. B. Turk-Browne. 2015. "Attention Stabilizes Representations in the Human Hippocampus." *Cerebral Cortex*, March, 1–14.
- Deuker, Lorena, Jacob L. S. Bellmund, Tobias Navarro Schröder, and Christian F. Doeller. 2016. "An Event Map of Memory Space in the Hippocampus." *eLife* 5 (October).
<https://doi.org/10.7554/eLife.16534>.
- Dudukovic, N. M., A. R. Preston, and J. J. Archie. 2011. "High-Resolution fMRI Reveals Match Enhancement and Attentional Modulation in the Human Medial Temporal Lobe." *Journal of Cognitive Neuroscience* 23 (3): 670–82.
- Eklund, Anders, Thomas E. Nichols, and Hans Knutsson. 2016. "Cluster Failure: Why fMRI Inferences for Spatial Extent Have Inflated False-Positive Rates." *Proceedings of the National Academy of Sciences*, June, 201602413–39.
- Floresco, Stan B., Christopher L. Todd, and Anthony A. Grace. 2001. "Glutamatergic Afferents from the Hippocampus to the Nucleus Accumbens Regulate Activity of Ventral Tegmental Area Dopamine Neurons." *The Journal of Neuroscience: The Official Journal of the Society for Neuroscience* 21 (13): 4915–22.
- Gasbarri, A., M. G. Packard, E. Campana, and C. Pacitti. 1994. "Anterograde and Retrograde Tracing of Projections from the Ventral Tegmental Area to the Hippocampal Formation in the Rat." *Brain Research Bulletin* 33 (4): 445–52.
- Gruber, Matthias J., Bernard D. Gelman, and Charan Ranganath. 2014. "States of Curiosity Modulate Hippocampus-Dependent Learning via the Dopaminergic Circuit." *Neuron* 84 (2): 486–96.
- Heuvel, Martijn P. van den, and Olaf Sporns. 2013. "Network Hubs in the Human Brain." *Trends in Cognitive Sciences* 17 (12): 683–96.

- Iordan, Marius Cătălin, Michelle R. Greene, Diane M. Beck, and Li Fei-Fei. 2016. "Typicality Sharpens Category Representations in Object-Selective Cortex." *NeuroImage* 134 (July): 170–79.
- Lisman, John E., and Anthony A. Grace. 2005. "The Hippocampal-VTA Loop: Controlling the Entry of Information into Long-Term Memory." *Neuron* 46 (5): 703–13.
- Lisman, John, Anthony A. Grace, and Emrah Düzel. 2011. "A neoHebbian Framework for Episodic Memory; Role of Dopamine-Dependent Late LTP." *Trends in Neurosciences* 34 (10): 536–47.
- Misaki, Masaya, Youn Kim, Peter A. Bandettini, and Nikolaus Kriegeskorte. 2010. "Comparison of Multivariate Classifiers and Response Normalizations for Pattern-Information fMRI." *NeuroImage* 53 (1): 103–18.
- Mumford, Jeanette A., Benjamin O. Turner, F. Gregory Ashby, and Russell A. Poldrack. 2012. "Deconvolving BOLD Activation in Event-Related Designs for Multivoxel Pattern Classification Analyses." *NeuroImage* 59 (3): 2636–43.
- Munn, Brandon R., Eli J. Müller, Gabriel Wainstein, and James M. Shine. 2021. "The Ascending Arousal System Shapes Neural Dynamics to Mediate Awareness of Cognitive States." *Nature Communications* 12 (1): 1–9.
- Nieoullon, André. 2002. "Dopamine and the Regulation of Cognition and Attention." *Progress in Neurobiology* 67 (1): 53–83.
- O'Craven, K. M., P. E. Downing, and N. Kanwisher. 1999. "fMRI Evidence for Objects as the Units of Attentional Selection." *Nature*, January.
<http://www.nature.com/nature/journal/v401/n6753/abs/401584a0.html>.
- Schlichting, Margaret L., Jeanette A. Mumford, and Alison R. Preston. 2015. "Learning-Related Representational Changes Reveal Dissociable Integration and Separation Signatures in the Hippocampus and Prefrontal Cortex." *Nature Communications* 6 (August): 8151.
- Schott, Björn H., Luciano Minuzzi, Ruth M. Krebs, David Elmenhorst, Markus Lang, Oliver H. Winz, Constanze I. Seidenbecher, et al. 2008. "Mesolimbic Functional Magnetic Resonance Imaging Activations during Reward Anticipation Correlate with Reward-Related Ventral Striatal Dopamine Release." *The Journal of Neuroscience: The Official Journal of the Society for Neuroscience* 28 (52): 14311–19.
- Semedo, João D., Amin Zandvakili, Christian K. Machens, Byron M. Yu, and Adam Kohn. 2019. "Cortical Areas Interact through a Communication Subspace." *Neuron* 102 (1): 249–59.e4.
- Shine, James M., Peter T. Bell, Elie Matar, Russell A. Poldrack, Simon J. G. Lewis, Glenda M. Halliday, and Claire O'Callaghan. 2019a. "Dopamine Depletion Alters Macroscopic Network

- Dynamics in Parkinson's Disease." *Brain: A Journal of Neurology* 142 (4): 1024–34.
- Shine, James M., Michael Breakspear, Peter T. Bell, Kayla Ehgoetz Martens, Richard Shine, Oluwasanmi Koyejo, Olaf Sporns, and Russell A. Poldrack. 2019b. "Human Cognition Involves the Dynamic Integration of Neural Activity and Neuromodulatory Systems." *Nature Neuroscience* 22 (2): 289–96.
- Swanson, L. W. 1982. "The Projections of the Ventral Tegmental Area and Adjacent Regions: A Combined Fluorescent Retrograde Tracer and Immunofluorescence Study in the Rat." *Brain Research Bulletin* 9 (1-6): 321–53.
- Thiele, Alexander, and Mark A. Bellgrove. 02/2018. "Neuromodulation of Attention." *Neuron* 97 (4): 769–85.
- Tsetsenis, Theodoros, Julia K. Badya, Julianne A. Wilson, Xiaowen Zhang, Elizabeth N. Krizman, Manivannan Subramanian, Kechun Yang, Steven A. Thomas, and John A. Dani. 2021. "Midbrain Dopaminergic Innervation of the Hippocampus Is Sufficient to Modulate Formation of Aversive Memories." *Proceedings of the National Academy of Sciences of the United States of America* 118 (40). <https://doi.org/10.1073/pnas.2111069118>.
- Urgolites, Zhisen J., John T. Wixted, Stephen D. Goldinger, Megan H. Papesh, David M. Treiman, Larry R. Squire, and Peter N. Steinmetz. 2020. "Spiking Activity in the Human Hippocampus prior to Encoding Predicts Subsequent Memory." *Proceedings of the National Academy of Sciences*, June, 202001338.
- Vishnu P Murty, Ian C Ballard, and R. Alison Adcock. 2016. "Hippocampus and Prefrontal Cortex Predict Temporally Distinct Timescales of Activation in Human Ventral Tegmental Area." *Cerebral Cortex*, in Press.
- Wade, Shirlene, and Celeste Kidd. 2019. "The Role of Prior Knowledge and Curiosity in Learning." *Psychonomic Bulletin & Review*, May. <https://doi.org/10.3758/s13423-019-01598-6>.
- Xue, G., Q. Dong, C. Chen, Z. Lu, J. A. Mumford, and R. A. Poldrack. 2010. "Greater Neural Pattern Similarity Across Repetitions Is Associated with Better Memory." *Science* 330 (6000): 97–101.
- Yoo, Seng Bum Michael, and Benjamin Y. Hayden. 2020. "The Transition from Evaluation to Selection Involves Neural Subspace Reorganization in Core Reward Regions." *Neuron* 105 (4): 712–24.e4.
- Zylberberg, Joel, Robert A. Hyde, and Ben W. Strowbridge. 2016. "Dynamics of Robust Pattern Separability in the Hippocampal Dentate Gyrus." *Hippocampus* 26 (5): 623–32.

REVIEWERS' COMMENTS

Reviewer #2 (Remarks to the Author):

I commend the authors for their thoughtful and detailed revision. I very much appreciate their including the litany of control analyses and methodological clarifications that I requested. I believe that these build confidence in the authors' interpretation of the findings. I do have a few remaining small points of clarification. To be clear, I don't think that any of these undermine any of the central claims, but include them for completeness.

2C "... Relatedly, I find that this procedure often yields a small fraction of estimates with extreme outlying values that can be quite influential in correlation analyses. It doesn't look like the authors removed or winsorized outliers. I would strongly recommend assessing whether the data contain outliers and identifying their impact."

the authors' did not respond to this component of my request, perhaps because my use of "this procedure" was quite vague! My recommendation to inspect outliers was not related to the normalization of t-values to z-values, but instead was related to obtaining single trial estimates for individual voxels in general. Since outliers can be quite problematic for multivariate analyses, I would still recommend that the authors inspect the resultant distributions of z-values to determine if they contain extreme values.

3E "did the authors guard against Simpson's Paradox and, more broadly, the contamination of within-subject analyses with between-subject variance? Common approaches include mean centering all predictors within-participant or including additional terms in the model to capture individual differences."

The authors' did not directly respond to the comment but instead emphasized that they used mixed models. Simpson's Paradox (or the confusion of within and between subject effects) is particularly problematic in mixed models, which is why I raised the concern. A simple solution is to center variables that show individual differences within participants in the model. The authors may have already taken this precaution, but I'm including a relevant methods paper to make sure that the issue is clear.

van de Pol, M., & Wright, J. (2009). A simple method for distinguishing within- versus between-subject effects using mixed models. *Animal Behaviour*, 77(3), 753–758.
<https://doi.org/10.1016/j.anbehav.2008.11.006>

7 "To control for spurious correlations, a null distribution was generated for each voxel using a permutation-based approach (500 iterations), and the r-value at the 95th percentile was subtracted from each voxel. This approach ensured that only correlation values greater than the 95th percentile of the null distribution are positive. A one-sample t-test was implemented using SPM12". I'm not familiar with this hybridization of non-parametric and parametric statistics. On the surface, it feels like an

overcorrection on results that yield few clusters. Could the authors provide more logic, or better yet precedence for this approach?

Here too I fear that my wording did not clearly articulate the issue as it focussed too much on the combination of non-parametric and parametric statistics. My concern is more so how they were combined. Take an example where all participants have high r-values in a voxel; half at the 94th percentile of the null distribution and half at the 96th percentile. In the authors' approach, this would yield an equal number of values just above and just below zero. These are then submitted to a one-sample t-test, which would yield insignificant findings. However, such consistently high r-values are highly unlikely to reflect a null effect. This is why I believe their approach of both thresholding at the 95th percentile and then running a t-test is overly conservative. Their alternative "While we could also use the raw (or fisher-z transformed) r-values at the individual level and perform permutation analysis at the group level (similar to that done in various packages with beta-values), this would similarly yield a hybridization of both non-parametric and parametric approaches." is a much better (and more common) solution and I have no qualms about this combination of parametric and non-parametric stats.

Pg. 22 – "Recent human work using depth electrodes showed hippocampal activity preceding successful encoding was unrelated to attention or arousal (Urgolites et al., 2020)." Throughout the response letter and revised paper, the authors used this citation to argue that pre-stimulus hippocampal activity is not modulated by attention. In my reading of this paper, however, the experiment doesn't measure or manipulate attention/arousal. Rather they infer that the observed subsequent memory effects don't reflect attention because parallel patterns are not seen before retrieval. Accordingly, I recommend that the authors use this citation more conservatively so as not to imply that attention or arousal were variables in the work.

Reviewer #3 (Remarks to the Author):

The authors have fully addressed all my concerns.

We thank the Reviewer for the clarifications and the thoughtful reading of our response. We have included point-by-point responses (in blue) to Reviewer #2. Amendments to the manuscript have been highlighted here in italics, for ease of review.

Reviewer #2 (Remarks to the Author):

I commend the authors for their thoughtful and detailed revision. I very much appreciate their including the litany of control analyses and methodological clarifications that I requested. I believe that these build confidence in the authors' interpretation of the findings. I do have a few remaining small points of clarification. To be clear, I don't think that any of these undermine any of the central claims, but include them for completeness.

2C "... Relatedly, I find that this procedure often yields a small fraction of estimates with extreme outlying values that can be quite influential in correlation analyses. It doesn't look like the authors removed or winsorized outliers. I would strongly recommend assessing whether the data contain outliers and identifying their impact."

the authors' did not respond to this component of my request, perhaps because my use of "this procedure" was quite vague! My recommendation to inspect outliers was not related to the normalization of t-values to z-values, but instead was related to obtaining single trial estimates for individual voxels in general. Since outliers can be quite problematic for multivariate analyses, I would still recommend that the authors inspect the resultant distributions of z-values to determine if they contain extreme values.

Response: We apologize for missing out on this comment in our prior response. As the Reviewer mentions, single trial estimates do indeed tend to be noisier, and while we did not explicitly remove any outlying values, several other measures were taken to account for this.

First, we opted for using the z-values instead of the beta values. The transformation of the beta value to the t-value (and then z) takes into account the standard error estimate for each voxel, and thus inherently downweigh noisy voxels. Second, our choice of correlation distance was also selected as a means of negating potential extreme values. While correlation distance is not immune to the effects of outliers, it is invariant to scaling, and as such, is less likely to be influenced by extreme values related to global shifts (in contrast to distance metrics such as Euclidean distance). Finally, in the analyses involving distance metrics, the mean amplitude of each ROI was also included as covariates to minimize potential effects driven by extreme values.

To alleviate further concerns, we repeated key analyses which involved the use of multivariate distances after removing trials with outlying values for each subject (trials with distance values exceeding 1.5 * interquartile range). A total of 15 outlying trials were identified across all participants (i.e. less than 1 per subject). After removing these trials, hippocampal typicality remained a significant predictor of memory (\$b = -.553\$, \$SE = .199\$, \$p = .005\$ ), and VTA activation remained a significant predictor of Hippocampal typicality (\$b = -.048\$, \$SE = .011\$, \$p = .0003\$ ). We have now included the following analysis in the supplementary results.

*Supplementary Results - "While including the mean amplitude as a covariate and using correlation distance measures reduces the influence of extreme values, we repeated key analyses after removing trials with outlying values for each subject (defined as trials with values exceeding 1.5 * interquartile range). A total of 15 outlying trials were identified, and the relationships remained significant between VTA activation and hippocampal typicality ($b = -.048$, $SE = .011$, $p = .0003$) and hippocampal typicality and memory ($b = -.553$, $SE = .199$, $p = .005$)."*

3E "did the authors guard against Simpson's Paradox and, more broadly, the contamination of within-subject analyses with between-subject variance? Common approaches include mean centering all predictors within-participant or including additional terms in the model to capture individual differences."

The authors' did not directly respond to the comment but instead emphasized that they used mixed models. Simpson's Paradox (or the confusion of within and between subject effects) is particularly problematic in mixed models, which is why I raised the concern. A simple solution is to center variables that show individual differences within participants in the model. The authors may have already taken this precaution, but I'm including a relevant methods paper to make sure that the issue is clear.

van de Pol, M., & Wright, J. (2009). A simple method for distinguishing within- versus between-subject effects using mixed models. *Animal Behaviour*, 77(3), 753–758.

<https://doi.org/10.1016/j.anbehav.2008.11.006>

Response: We thank the reviewer for the clarification, and we now have a better understanding of the Reviewer's original comment. Based on the suggestions, we conducted additional control analyses for our key findings. As recommended in van De Pol & Wright (2009), we performed the analyses using the within-subject centered regressor in conjunction with a regressor for the subject mean. Anticipatory VTA activation remained a significant predictor of Memory ($b = .101$, $SE = .041$, $p = .015$), hippocampal typicality remained a significant predictor of memory ($b = -.556$, $SE = .197$, $p = .005$), and VTA activation remained a significant predictor of hippocampal patterns ($b = -.049$, $SE = .012$, $p = .0003$). We hope these additional analyses would provide greater confidence in our overall findings.

7 "To control for spurious correlations, a null distribution was generated for each voxel using a permutation-based approach (500 iterations), and the r-value at the 95th percentile was subtracted from each voxel. This approach ensured that only correlation values greater than the 95th percentile of the null distribution are positive. A one-sample t-test was implemented using SPM12". I'm not familiar with this hybridization of non-parametric and parametric statistics. On the surface, it feels like an overcorrection on results that yield few clusters. Could the authors provide more logic, or better yet precedence for this approach?

Here too I fear that my wording did not clearly articulate the issue as it focussed too much on the combination of non-parametric and parametric statistics. My concern is more so how they were combined. Take an example where all participants have high r-values in a voxel; half at the 94th percentile of the null distribution and half at the 96th percentile. In the authors' approach,

this would yield an equal number of values just above and just below zero. These are then submitted to a one-sample t-test, which would yield insignificant findings. However, such consistently high r-values are highly unlikely to reflect a null effect. This is why I believe their approach of both thresholding at the 95th percentile and then running a t-test is overly conservative. Their alternative “While we could also use the raw (or fisher-z transformed) r-values at the individual level and perform permutation analysis at the group level (similar to that done in various packages with beta-values), this would similarly yield a hybridization of both non-parametric and parametric approaches.” is a much better (and more common) solution and I have no qualms about this combination of parametric and non-parametric stats.

Response: We thank the reviewer for the clarification, and while we agree that our approach may be conservative, we think that this is appropriate for an exploratory search, and we do not think that permutation at the group-level provides a simple solution. In our prior response, we highlighted potential concerns with permuting at the group-level, and also stated our rationale for permuting at the subject-level along with similar precedence of corrections at both levels (e.g. Schlichting, Mumford, and Preston 2015; Deuker et al. 2016). Based on the above, we believe that our current approach is justified, and we hope the reviewer agrees with the soundness of our approach.

Pg. 22 – “Recent human work using depth electrodes showed hippocampal activity preceding successful encoding was unrelated to attention or arousal (Urgolites et al., 2020).” Throughout the response letter and revised paper, the authors used this citation to argue that pre-stimulus hippocampal activity is not modulated by attention. In my reading of this paper, however, the experiment doesn’t measure or manipulate attention/arousal. Rather they infer that the observed subsequent memory effects don’t reflect attention because parallel patterns are not seen before retrieval. Accordingly, I recommend that the authors use this citation more conservatively so as not to imply that attention or arousal were variables in the work.

Response: As the Reviewer mentions, Urgolites et al do not directly manipulate attention/arousal, but instead drew their inference from the absence of a predictive effect on concurrent task performance. We agree with the reviewer that this should be more clearly specified, and we have amended our use of this citation accordingly.

Page 24 - “Recent human work using depth electrodes showed that pre-stimulus hippocampal spiking predicted subsequent memory but not ongoing retrieval accuracy, suggesting that hippocampal activity preceding successful encoding is unlikely to be related to attention or arousal”